# Measurement report: Underestimated reactive organic gases from residential combustion: insights from a near-complete speciation

Yaqin Gao[1], Hongli Wang[1], Lingling Yuan[1], Shengao Jing[1], Bin Yuan[2], Guofeng Shen[3], Liang Zhu[4,5], Abigail Koss[5], Yingjie Li[1], Qian Wang[1], Dan Dan Huang[1], Shuhui Zhu[1], Shikang Tao[1], Shengrong Lou[1], Cheng Huang[1]

[1]1State Environmental Protection Key Laboratory of Formation and Prevention of Urban Air Pollution Complex, Shanghai Academy of Environmental Sciences, Shanghai 200233, China
[2]Institute for Environmental and Climate Research, Jinan University, Guangzhou 511443, China
[3]Laboratory of Earth Surface Processes, College of Urban and Environmental Science, Peking University, Beijing 100871, China
[4]Tofwerk China, Nanjing 210000, China
[5]Tofwerk AG, Thun 3645, Switzerland

*Correspondence to*: Hongli Wang (wanghl@saes.sh.cn); Cheng Huang (huangc@saes.sh.cn)

**Abstract.** Reactive organic gases (ROGs), as important precursors of secondary pollutants, are not well resolved as the chemical complexity challenged its quantification in many studies. Here, a near-complete speciation of ROG emissions from residential combustion was developed by the combination of Proton Transfer Reaction Time-of-Flight Mass Spectrometer (PTR-ToF-MS) with Gas Chromatography equipped with a Mass Spectrometer and a Flame Ionization Detector (GC-MS/FID), including 1049 species in all. Among them, 125 identified species, ~90% of the total ROG masses, were applied to evaluate their emission characteristics through the real combustion samplings in rural households of China. The study revealed that with 55 species mainly oxygenated species, higher hydrocarbons with $\geq 8$ carbon atoms, and nitrogen-containing ones, previously un- and under-characterized, ROG emissions from residential coal and biomass combustion were underestimated by $44.3\% \pm 11.8\%$ and $22.7\% \pm 3.9\%$, respectively, which further amplified the underestimation of secondary organic aerosols formation potential (SOAP) as high as $70.3\% \pm 1.6\%$ and $89.2\% \pm 1.0\%$, respectively. The hydroxyl radical reactivity (OHR) of ROG emissions was also undervalued significantly. The study provided a feasible method for the near-complete speciation of ROGs in atmosphere and highlighted the importance of acquiring completely speciated measurement of ROGs from residential emissions, as well as other processes.

**Highlights**

A near-complete speciation of ROG emissions from residential combustion was developed.

Oxygenated species, higher hydrocarbons and nitrogen-containing ones played larger roles in the emissions of residential combustion compared with the common hydrocarbons.

ROG emissions from residential combustion were largely underestimated, leading more underestimation of their OHR and SOAP.

**Keywords:** Chemical speciation, Reactive organic gases, Residential combustion, Estimation bias

## 1 Introduction

Residential combustion, dominated by approximately 89% solid fuels in China (Zhu et al., 2019), is responsible for ~23% and ~71% of the outdoor and indoor $PM_{2.5}$ concentrations, and ~67% of $PM_{2.5}$-induced premature deaths (Yun et al., 2020). Reactive organic gases (ROGs), organic gases other than methane, from residential combustion, have been shown to serve as key precursors for secondary organic aerosols (SOA) (Huo et al., 2021b) and ozone formation (Heald and Kroll, 2020; Heald et al., 2020). ROG emissions from residential combustion especially biomass combustion have been widely studied due to their great contribution to global ROGs and the complexity of compositions. Among them, studies focusing on the ROG speciation for residential combustion could generally be divided into three categories according to the measurement methods, as listed in Table S1. The first one was the whole-air sampling with offline analysis by one-dimensional gas chromatography system equipped with a mass spectrometer and/or a flame ionization detector (GC-MS/FID), which mainly focused on the hydrocarbons (<C12) (Mo et al., 2016; Wang et al., 2013; Liu et al., 2008). With the development of the advanced instruments, the second category of studies on ROG emissions gave more attention to the polar species like oxygenated ROGs, which could be online detected through the whole combustion process mainly by proton transfer reaction time-of-flight mass spectrometry ($H_3O^+$ PTR-ToF-MS) due to the high mass resolution and sensitivity (Cai et al., 2019; Bruns et al., 2017; Stockwell et al., 2015; Koss et al., 2018; Akherati et al., 2020; Wu et al., 2022). Considerable (approximately 6%-24%) species with intermediate volatility in residential ROG emissions were identified as the large contributors of SOA (Cai et al., 2019; Koss et al., 2018).

Thirdly, due to the inability to isolate isomers by $H_3O^+$ PTR-ToF-MS and considerable amount of oxygenated ROGs with intermediate volatility in residential ROG emissions, the increasing interest has been put on the application and comparisons of multiple instruments for the detailed identification of ROG species (Koss et al., 2018; Hatch et al., 2017). More than 150 PTR ion masses were identified using the combination of techniques including GC pre-separation, two-dimensional GC system (GC×GC), fourier transform infrared spectroscopy (FTIR), and $NO^+$ chemical ionization mass spectrometer ($NO^+$ CIMS), which contributed ~90% of the ROG masses detected by $H_3O^+$ PTR-ToF-MS in biomass combustion emissions (Koss et al., 2018). The comparisons demonstrated that $H_3O^+$ PTR-ToF-MS might be the most suitable for the detection of the lowest-volatility and most polar species, which covered the most (50%-79%) species, comparing with the other instruments, in the combined ROG measurement covering more than 500 species from different instruments (Koss et al., 2018; Hatch et al., 2017). Recently, the higher alkanes ($\geq$C8), one kind of considerable species in residential combustion emissions (Jathar et al., 2014; Huo et al., 2021a; Li et al., 2023), which were not included in the comprehensive measurements of previous studies (Hatch et al., 2017), could be well measured by PTR-ToF-MS with $NO^+$ ion chemistry (Wang et al., 2020a; Koss et al., 2016). Thus, PTR-ToF-MS might be a preferent and promising method for the development of near-complete ROG speciation relevant for residential combustion, but need to combine with GC-MS/FID for the complement measurement of aliphatic hydrocarbons. Therefore, the present study focused on (1) developing the near-complete ROG speciation through quantifying all signals by $H_3O^+$ PTR-ToF-MS and supplementing C2-C22 aliphatic hydrocarbons by GC-MS/FID and $NO^+$ PTR-ToF-MS, and (2) the

composition of ROG emissions through the real combustion sampling in rural household of China. Finally, the near-complete ROG speciation further supported the estimation of the ROG emissions from residential combustion in China as well as their

hydroxy radical reactivity and formation potential of SOA. The present study took the residential combustion as an example for developing the near-complete ROG speciation mainly considering the large complexity of combustion-relevant ROG speciation and the comprehensive measurement of residential combustion previously, which could be used to further confirm the present result by overlapping species.

## Materials and methods

**2.1 Sampling**

The ROG samples of the combustion of four typical biomass fuels (wood, corncob, bean straw and corn straw) and two typical coals (anthracite, and briquette coals) (see Fig. S1 in the Supplementary Information) were collected from the stack nozzles of household stoves by vacuumed SUMMA canisters (Entech Inc., 3.2 L). During canister sampling, the combustion in the stoves was in the stage of flaming visually, which was the common condition for heating or cooking in the rural areas in northern

China. Particles were removed by a particle-filter with 5.0 μm pore size Teflon filter (PTFE). Temperatures ($37 \pm 17$ °C) of flue gas were monitored at the sampling location by a flue gas analyzer (Testo 350). Wood and straws burned at an average temperature of 394 °C (in the range of 231-567 °C) and 353 °C (334-371 °C), respectively, while the residential coals burned at a higher temperature (514 °C, 411-581 °C), which were measured by an infrared thermometer in the stove. Here totally 23 samples were collected, as shown in Fig. S1. All the samples stored in SUMMA canisters were detected with the combination

of $H_3O^+/NO^+$ PTR-ToF-MS and GC-MS/FID within 9 days.

**2.2 Analysis by $H_3O^+$ PTR-TOF**

$H_3O^+$ PTR-ToF-MS (Vocus 2R, Tofwerk AG, Switzerland) served as the primary equipment toward ROG complete speciation due to the detectability of most ROGs with relatively complete species coverage (Li et al., 2020; Krechmer et al., 2018) and theoretically computable sensitivity (Sekimoto et al., 2017). All valid signals in raw mass spectrum were identified, quantified,

and analyzed for uncertainty, following the detailed process of data treatment shown in Fig. 1.

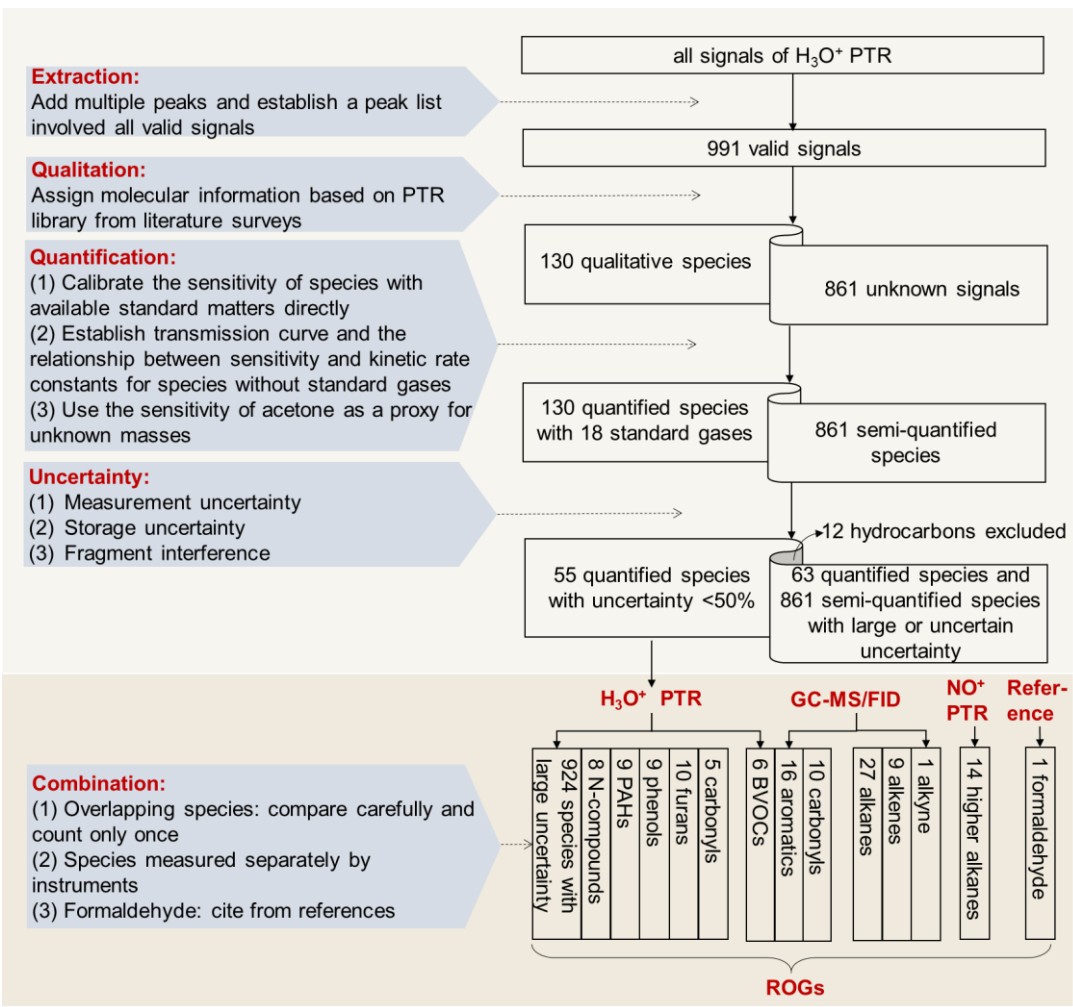

**Figure 1.** Diagram of data treatment to obtain the near-complete ROG speciation.

### 2.2.1 Identification

A total of 991 valid ion masses were extracted from original mass spectrum through multi-peak fitting (Stark et al., 2015), of which 130 species were well-qualitative while 861 signals were still unknown according to literature surveys (Pagonis et al., 2019; Koss et al., 2018; Stockwell et al., 2015).

**Extraction (991 masses).** The raw mass spectrum data of gaseous ROGs measured by $H_3O^+$ PTR-ToF-MS were mainly distributed in the range of m/z 40-m/z 200 (Koss et al., 2017), with a mass resolution of 10000 m $\Delta$ m$^{-1}$. Multiple peaks were
added at each nominal mass below m/z 200 resulting in a peak list of 991 valid masses in the software Tofware package v3.2.3 (Tofwerk Inc.).

**Qualitation (130 specific ROGs and 861 unknown masses).** Based on accurate m/z, isotope pattern and published PTR library from the review of 49 publications (containing 226 molecules and nearly 1000 species) (Pagonis et al., 2019) and local

PTR library (Yuan et al., 2022; Gao et al., 2022), molecular information of individual mass was calculated and matched. Due to the complexity of isomers, the recommended names of various species were further determined by the combustion-relevant reports of Koss et al. (2018) and Stockwell et al. (2015) (Koss et al., 2018; Stockwell et al., 2015). Finally, 130 masses matched with proposed species, while the remaining 861 peaks were defined as unknown masses with the molecular formula but without recommend species or only with accurate m/z without elemental composition calculated.

### 2.2.2 Quantification

Sensitivities of all the masses in this work were obtained by (1) calibration by authentic standard for species with standard matters available, (2) theoretical calculation by the relationship between sensitivity and kinetic rate constants ($k$) (Sekimoto et al., 2017) for identified species without commercial standards, or (3) estimation using the sensitivity of acetone as a proxy for unknown masses.

**Calibration method (18 species).** The sensitivities of 18 species were calibrated experimentally by introducing multi-gradient known concentrations of given ROGs from a customized cylinder (Linde Gas North America LLC, USA). These standard gases included benzene, toluene, styrene, m-xylene, 1,3,5-trimethylbenzene, m-diethylbenzene, isoprene, α-pinene, acetaldehyde, acrolein, acetone, methyl vinyl ketone, methyl ethyl ketone, 2-pentanone, methyl isobutyl ketone, acetonitrile, furan, and naphthalene. More details of the calibration could be found in our previous studies (Gao et al., 2022; Wang et al., 2020b).

**Calculation method (112 species).** For other well qualitative species without available standards, we used the method proposed by Sekimoto et al. (2017) to theoretically calculate the concentrations by the relationship between ROG sensitivities and kinetic rate constants ($k_{kinetic}$) for proton transfer reactions of $H_3O^+$ with ROGs (Fig. S2), which was established by the calibrated species above.

**Estimation method (861 masses).** For unknown masses, it is difficult to estimate $k_{kinetic}$ which depends on molecular properties and the sensitivity of acetone was used as a proxy (Cai et al., 2019).

### 2.2.3 Uncertainty analysis

The measurement uncertainty of a specific ROG is mainly from the analysis and storage. The analysis uncertainty depending on the quantification method in Section 2.2.2 was estimated following the method of Sekimoto et al. (2017). The effect of storage inside the canister was evaluated using the standard samples with different storage duration in the laboratory. Accordingly, the total measurement uncertainties of different ROG species were estimated, as listed in Table S2 in the Supplementary Information.

**Analysis uncertainty.** The analysis uncertainty corresponds to the three kinds of quantification methods. Firstly, the experimentally calibrated species have an upper-bound analysis uncertainty of 15%, including those of analysis precision and calibration factors (Gao et al., 2022), as shown in Table S2. Secondly, for those species quantified by theoretical calculation, the uncertainty was mainly attributed to the estimated value of $k_{kinetic}$ and the linear regression slope (~20% from fitting bias

in this study) between sensitivity and $k_{kinetic}$, which was within 50% (Sekimoto et al., 2017). Thirdly, for unknown masses, the uncertainty was mainly from the difference of their "real" sensitivities from the assumed sensitivity of acetone, which was dependent on the difference of their $k_{kinetic}$. Generally, the oxygenated species ($C_xH_yO_z$) have higher (~2.5-3.5×10$^{-9}$ cm$^3$ mol$^{-1}$ s$^{-1}$) $k_{kinetic}$ than those of aliphatic hydrocarbons ($C_xH_y$, ~2×10$^{-9}$ cm$^3$ mol$^{-1}$ s$^{-1}$), and the $k_{kinetic}$ of acetone is 3.23×10$^{-9}$ cm$^3$ mol$^{-1}$

s$^{-1}$ (Cappellin et al., 2012). Thus, the maximum uncertainty of the estimated concentrations of unknown masses caused by this assumption is estimated to be 65% although the "real" $k_{kinetic}$ is as low as 2×10$^{-9}$ cm$^3$ mol$^{-1}$ s$^{-1}$ in this study.

**Loss in storage.** To evaluate the possible artifacts caused by the SUMMA canister storage, the sensitive tests of the storage duration were carried out by the standard samples in laboratory. Standard samples including 85 species, with a known concentration (5 ppbv in this study) were prepared into clean vacuum SUMMA canisters and detected within 2h as well as on

days 1, 2, 4, 7, 10 and 14 after preparation. Here, the loss fraction was estimated to be the maximum threshold because the mixing ratios of most observed ROGs in all samples were normally higher than 5 ppbv. The detailed species and deviations were shown in Fig. S3. The storage loss on day10$^{th}$ was used considering that samples were analyzed within 9 days after sampling in this work (Table S2). According to experimental results, the loss of some carbonyls (-CHO and RCO-), furans and nitrogenous species (-CN) was measured within 20%, while the loss of several alcohols (-OH) exceeded 50%. The results

of carbonyls were comparable with previous studies in which the half-lives of aldehydes were 18 days in the canister (Batterman et al., 1998). The higher polarity of alcohols is a possible reason for their larger loss proportion, as the polar species preferentially adsorbed on surface sorption sites of SUMMA canister inner walls (Batterman et al., 1998). Although volatility is another potential factor, in current experimental results for 10 days, the loss proportion has no significant dependence on volatility (Fig. S4).

For the other species of carbonyls, furans and nitrogenous species without standards, their loss during storage was assumed as the measured largest loss fraction in the same category. For phenols, an important category in ROG emissions from residential combustion (Bruns et al., 2016), their loss was believed to be predictable considering that phenols have been mentioned suitable for canister sampling in US EPA method TO15 (Epa/625/R-96/010b, 1999) and was estimated as 20% referring to the largest loss fraction of measured carbonyls.

For other categories such as acids (-COOH), species with more than 2 oxygen atoms (usually 2 functional groups) and components containing -NO$_n$, there were no standards used to evaluate the loss during storage. Overall, 63 specific ROGs with large uncertainty potentially (>50%) and 861 unknown masses with uncertain loss fraction were only used to quantify the total ROGs and were excluded from the further discussion of ROG composition.

**Fragment interference.** The aliphatic hydrocarbons such as $C_3H_6H^+$, $C_4H_8H^+$ are interfered by fragments from different

species. In this work, 12 such ions were excluded from the results to avoid the uncertainty, due to well-characterized hydrocarbons were provided by GC-MS/FID.

In summary, except for 12 aliphatic hydrocarbon fragments, the other 979 detected masses by $H_3O^+$ PTR-ToF-MS were used in this study, among which 55 species with the uncertainty ranging from 1% to 44% were used in further discussion of ROG

composition, and the rest masses with larger (>50%) or unknown uncertainty potentially were only used to quantify the total

ROGs in Fig. 2.

### 2.2.4 Limitation

$H_3O^+$ PTR-ToF-MS could identify the ROGs as long as the proton affinity of ROGs greater than that of water (691 kJ mol$^{-1}$) (Yuan et al., 2017), with relatively complete species coverage. It may be the preferred method toward ROG complete measurement, because most ROGs can be detected (Li et al., 2020; Krechmer et al., 2018) and the sensitivity for a given ROG

can be calculated theoretically by $H_3O^+$ PTR-ToF-MS (Sekimoto et al., 2017). Despite these advantages, $H_3O^+$ PTR-ToF-MS has two limitations related to the reagent ion chemistry. Firstly, the technique is insensitive to C2-C7 alkanes, ethene and acetylene with lower proton affinity than water (Jobson et al., 2009). Secondly, higher alkanes (≥C8), one kinds of important components of the fuel combustion (Huo et al., 2021a; Jathar et al., 2014), were difficult to quantify by $H_3O^+$ PTR-ToF-MS due to fragments produced during the ionization process.

Formaldehyde, a special species during PTR measurement was not detected in this study, suffering from double effects by (1) mass discrimination and (2) reversible reaction. Firstly, for the PTR-ToF-MS deployed in this study, as the transmission curve (Fig. S2) shows, the mass transmission efficiency of protonated formaldehyde at m/z 31 is close to 0, leading to an almost negligible sensitivity of formaldehyde. The decisive factor for transmission efficiency of PTR-ToF-MS is the radio frequency (RF) amplitude and the frequency of the big segmented quadrupole (BSQ), which can be set up as an ion filter (Wang et al.,

2020c). A similar situation has been reported by Yang et al. (2022), in which formaldehyde was not measured by the PTR-ToF-MS (Vocus 2R, Aerodyne Research Inc.) (Yang et al., 2022). Secondly, non-negligible back reactions between protonated formaldehyde and water vapor can reduce the sensitivity for formaldehyde (Spanel and Smith, 2008; Cui et al., 2016). For the developed Vocus instrument deployed in current work, the water vapor flow increases from the previous level 4-8 sccm (cm$^3$ min$^{-1}$ at 105 Pa and 273.15 K) (De Gouw et al., 2004) to 20-30 sccm (Krechmer et al., 2018). This change brings a great

advantage that the humidity dependence of instrument response to most species can be ignored. However, for formaldehyde, the reduction of its sensitivity is predictable because the really high water vapor concentration was provided for the back reactions.

### 2.3 Species combination

Aiming to develop a near-complete speciation of ROGs in this study, $NO^+$ PTR-ToF-MS (Vocus 2R, Tofwerk AG,

Switzerland), and GC-MS/FID (TH-300, Wuhan Tianhong Instruments, China) were deployed additionally against the limitations of $H_3O^+$ PTR-ToF-MS discussed above, which was highly complementary, with covering some unique and important range of compositional space. During the species combination for a near-complete speciation of ROGs, overlapping measurements of the same species should be counted only once.

### 2.3.1 NO[+] PTR-ToF-MS

NO[+] PTR-ToF-MS has been demonstrated to provide a supplementary measurement of higher alkanes (Wang et al., 2020a; Koss et al., 2016). Different from $H_3O^+$ PTR-ToF-MS by which the sensitivity for a given ROG can be calculated theoretically even without the standard for calibration, authentic standards are necessary for quantification by NO[+] PTR-ToF-MS, which limited the characterization of mass spectrum ionized by NO[+]. The difficulty to predict the ionized ROG products and to interpret the mass spectrum unambiguously further limited its application in ROG speciation, because NO[+] has three common

reaction mechanisms with ROGs: charge transfer, hydride abstraction, and cluster formation (Koss et al., 2016). Therefore, NO[+] PTR-ToF-MS in this study was only used for a supplementary measurement of higher alkanes (≥C8) with a well-established quantitative method (Wang et al., 2020a).

C8-C15 alkanes were calibrated using a custom cylinder (Linde Gas North America LLC, USA) and sensitivities of C16-C21 alkanes were assumed to be the same as that of C15 n-alkane according to Wang et al. (2022) (Wang et al., 2020a). The error

caused by this assumption was considered to be minimal, because the degree of fragmentation, a parameter inversely proportional to the sensitivity of higher alkanes, was similar between C16-C21 alkanes and C15 alkanes (~20%) (Wang et al., 2020).

### 2.3.2 GC-MS/FID

Against the limitations of PTR technologies (Arnold et al., 1998; Gueneron et al., 2015), GC-MS/FID with a cryogen-free

preconcentration device was also deployed. Fifty-seven hydrocarbons and 12 oxygenated organic compounds were measured by GC-MS/FID and quantified using gas standards (Linde gas North America LLC, USA).

### 2.3.3 Overlapping species

Thirty-two species measured by GC-MS/FID overlapped the 14 protonated ions measured by $H_3O^+$ PTR-ToF-MS (the pink shadow bar in Fig. 2), including 16 aromatic hydrocarbons, 10 carbonyls and 6 Biogenic Volatile Organic Compounds

(BVOCs), denoting the species generally emitted and formatted from natural sources in atmosphere. The concentrations of overlapping species in all samples were carefully compared and showed good consistencies between two instruments (slope = 1.00 ± 0.15, 0.94 < r < 0.99) (Fig. S5). Considering the better performance of GC-MS/FID in isomer differentiation, GC-MS/FID data were given precedence.

Although C8-C10 alkanes were measured by both GC-MS/FID and NO[+] PTR-ToF-MS, they represent different meanings.

The concentration of higher alkanes from NO[+] PTR-ToF-MS should be regarded as the summed concentrations of n-alkanes and branched alkanes that have the same chemical formulas. C8-C10 alkanes measured by GC-MS/FID were mainly normal alkanes and few branched alkanes. Following the principle of maximizing the species conservation, the GC data was used to preserve isomer speciation. Meanwhile, after subtracting the concentrations of relevant GC species, the NO[+] PTR-ToF-MS data was also remained as an independent species.

## 2.3.4 Formaldehyde

Formaldehyde, a very important species from combustion (Zarzana et al., 2017) but not measured in this study (Section 2.2.4), was assumed to account for ~7% and ~5% in all discussed species for coal and biomass combustion in this study, respectively, according to the previous reported results (Cai et al., 2019; Stockwell et al., 2015; Stockwell et al., 2016). To include the contribution of formaldehyde in the near-complete speciation of ROGs, the emission ratio of formaldehyde with the reference species in emissions was effective for this purpose. Generally, all the overlapping species measured in this and the previous studies could be used as the reference species, because the relative contribution of all the overlapping species agreed well between the current study and the previous studies as presented in Fig. S6 in the Supplementary Information. Benzene was chosen for normalization because in the emissions of all types of fuel combustion, benzene was the most abundant aromatics that were the major overlapping species between the current and previous studies. Specifically, the emission ratio of formaldehyde ($ER_{HCHO, ref}$) to benzene from anthracite coal (2.13 g/g, benzene) and straw (2.85 g/g, benzene) combustion could be obtained from Cai et al. (2019) and Stockwell et al. (2016). Considering the emission ratios of the species covered by both previous studies and the current work were in good agreement as shown in Fig. S6, the mass fraction of formaldehyde ($f_{HCHO,cal}$) in ROG emissions from combustion of this work could be calculated using the previously reported $ER_{HCHO, ref}$ and the currently measured mass fraction of benzene ($f_{benzene}$) as follows:

$$f_{HCHO,cal} = \frac{ER_{HCHO,ref} \times f_{benzene}}{1 + ER_{HCHO,ref} \times f_{benzene}}, \tag{1}$$

Finally, all the species included in this study were sketched in Fig. 2, as well as their mass fractions in ROG emissions from residential combustion which would be discussed in detail below. Combining 965 unique masses detected by $H_3O^+$ PTR-ToF-MS with 69 species detected by GC-MS/FID, 14 higher alkanes detected by $NO^+$ PTR-ToF-MS, as well as formaldehyde from references, a total of 1049 species were used in this study, including 27 alkanes, 9 alkenes, 1 alkyne, 16 aromatics, 6 BVOCs, 9 Polycyclic Aromatic Hydrocarbons (PAHs), 14 higher alkanes, 16 carbonyls, 10 furans (furan and its homologues and derivatives), 9 phenols (phenol and its homologues and derivatives), 8 nitrogen-containing species and 924 other species with large uncertainty. Among them, only 125 species with uncertainty below 50% were included in the speciation of ROGs from residential combustion emissions. Table S2 in the Supplementary Information listed the 125 species and their uncertainties. They contributed 89% ± 20% and 92% ± 34% of the total ROGs from residential combustion. More details of each kind of samples were shown in Fig. S7.

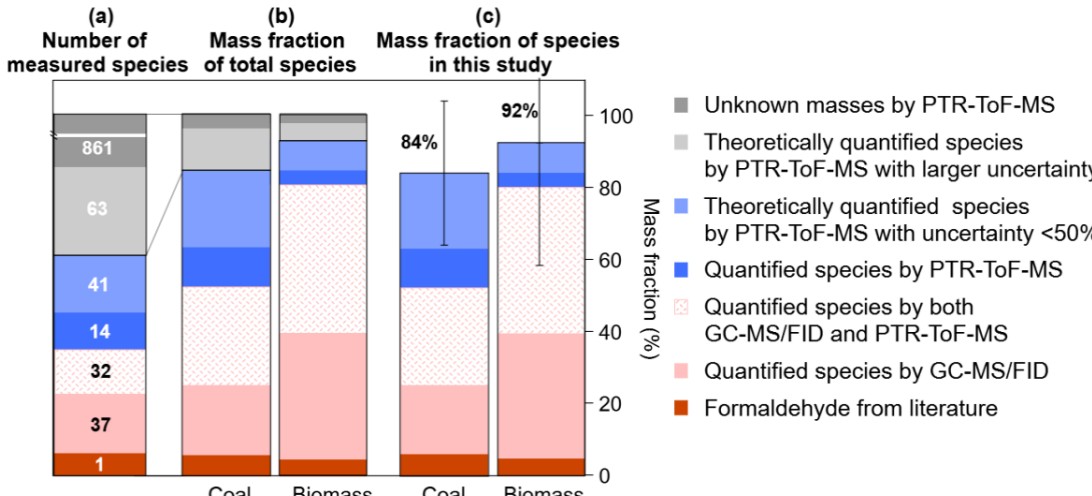

**Figure 2.** The number and mass fraction of species measured in this study. (a) The number of measured species. (b) The average mass fraction of all measured species from coal combustion and biomass combustion. (c) The mass fraction of selected 125 species in ROGs measured in this study. The result of formaldehyde was cited from previous studies (Cai et al., 2019; Stockwell et al., 2015). All the quantified species were calculated by standard matters. The theoretically quantified species were measured by $H_3O^+$ PTR-ToF-MS by the calculation method proposed by Sekimoto et al. (2017) to determine the relationship between ROG sensitivity and kinetic rate constants for proton transfer reactions of $H_3O^+$ with ROGs. Among the above theoretically quantified species, species like alcohols and acids (-OH and -COOH), species with more than 2 oxygen atoms (usually 2 functional groups) and components containing $-NO_n$, had relatively large uncertainty due to the potential loss in the storage. The unknown species were detected by $H_3O^+$ PTR-ToF-MS and were semi-quantified by the sensitivity of acetone.

## 3 Results and discussions

### 3.1 A near-complete speciation of ROGs from residential combustion

ROG compositions emitted from typical residential combustion using two types of coals (anthracite, and briquettes) and four types of biomasses (wood, corncob, corn straw and bean straw) are shown in Fig. 3. The measured ROG profiles for each kind of fuel had a good correlation (R > 0.6) (Fig. S8), and the average result was used here.

Generally, ROGs emitted from the residential combustion can be divided into three groups based on the element composition, including hydrocarbons, oxygenated species, and nitrogen-containing species. Differing from the previously studies which mainly stressed the dominant role of hydrocarbons (Mo et al., 2016; Stockwell et al., 2015; Wang et al., 2014), the contribution of oxygenated species (36.8%-56.8%) was comparable with that of hydrocarbons (40.8%-48.7%) in this study. It was expected that 24 more oxygenated species mainly including furans, phenols and carbonyls were measured by $H_3O^+$ PTR-ToF-MS in our

study, which were un- and under-characterized in previous studies using GC methods. Besides hydrocarbons and oxygenated species, nitrogen-containing species mainly included acetonitrile and acrylonitrile also played a considerable role in ROG emissions from residential combustion, with the proportions ranging from 5.7% to 14.5%, which have been previously reported (Cai et al., 2019).

Here, we defined the species previously un- and under-characterized by GC methods as newly identified species and could be measured only using $H_3O^+/NO^+$ PTR-ToF-MS in this study, and as a result 55 of 125 species were newly identified species. As shown in Fig. 3, these newly identified species mainly including furans and phenols contributed 44.3% ± 11.8% of the total ROGs for coal emissions and 22.7% ± 3.9% of the total ROGs for biomass emissions. We also compared our results with the previous reports, and for comparison the previous speciation was scaled by the total fraction of the previously reported species in the total ROGs measured in this study. As shown in Fig. 3 and Fig. S6, the fraction of reported species in previous studies was comparable to the present result. In particular, as Fig. 3(c) shows, the present composition of ROGs from residential wood combustion was close to that of Black Spruce combustion simulated in laboratory by multiple advanced trace-gas instruments, which reported 464-551 species (~173 molecules) in all (Hatch et al., 2017). It further confirmed that the obtained ROG characterization with the combination of $H_3O^+/NO^+$ PTR-ToF-MS and GC-MS/FID was nearly complete. Our study underscored the importance of the completely speciated measurement of the ROG emissions from residential combustion especially for coal combustion.

Large difference was observed between the ROG speciation of coal and biomass combustion but not significant among different types of coal or biomass, as shown in Fig. 3. Specifically, the alkenes mainly ethylene and propene dominated hydrocarbons emitted from biomass combustion, while alkanes were the most hydrocarbons from coal combustion. Especially, coal combustion emitted considerable higher alkanes including 8-21 carbon atoms and gaseous PAHs (mainly including 2-3 benzene rings), primarily generated by pyrolysis of the volatile matter in coal (Du et al., 2020), accounting for 8.3%-14.8% of ROGs much higher than the minor fractions (0.4%-2.4%) for biomass combustion emissions. In terms of oxygenated species, coal combustion emitted considerable furans (16.8% ± 3.2%) and phenols (6.1% ± 1.5%) mainly formed through pyrolysis of polymers in coal (Liu et al., 2017b; Morgan and Kandiyoti, 2014), which together played a comparable role with carbonyls (26.9% ± 6.8%) in ROGs. In comparison, carbonyls (40.6% ± 6.6%) were the dominant oxygenated species in ROG emissions from biomass combustion, mainly originated from products of biomass pyrolysis and pyrosynthesis (Morgan and Kandiyoti, 2014). A slightly higher proportion of phenols and furans from wood combustion (12.8% ± 3.0%) than straw combustion (8.7% ± 2.8%) were observed, possibly resulting from the higher composition of lignin in wood (Collard and Blin, 2014). Considerable terpenes were also observed in ROG emissions from residential coal (1.5% ± 0.2%) and biomass (3.2% ± 0.5%) combustion.

The ROG composition and individual species proportion in source profiles obtained in the present study were mainly from the measurements during the flaming stage. Considering the difference between flaming stage and the whole combustion cycle, the potential bias of the present results should be further discussed. By re-analyzing the data obtained from the authors (Cai et al., 2019) , the ROG composition from coal combustion in flaming stage and the whole cycle agreed well, which was expected

due to the small changes of ROG composition throughout the first three stages which emitted 96% of ROGs (Fig. S9). Similar results (Fig. S10) could be concluded from the re-analysis of the reported emission data from biomass combustion by Koss et al. (2018) (Koss et al., 2018). Furthermore, the proportion of individual species between the flaming stage and the whole cycle has a deviation in the range of -50% to 22% for coal and straw combustion (Fig. S11). Actually, the previous study of Gilman et al. (2015) have carefully compared discrete emission ratios (ERs) during flaming and smoldering combustion and fire-integrated ERs of the whole cycle and the average slope and standard deviation of discrete versus fire-integrated ERs for select ROGs from 56 biomass burns in the US was 1.2 ± 0.2 (Gilman et al., 2015). In summary, the bias of the fractions of species categorized by functional group from both coal and biomass combustion obtained in our study was negligible, and the bias of individual species proportion from both coal and biomass was estimated to be within 50% generally.

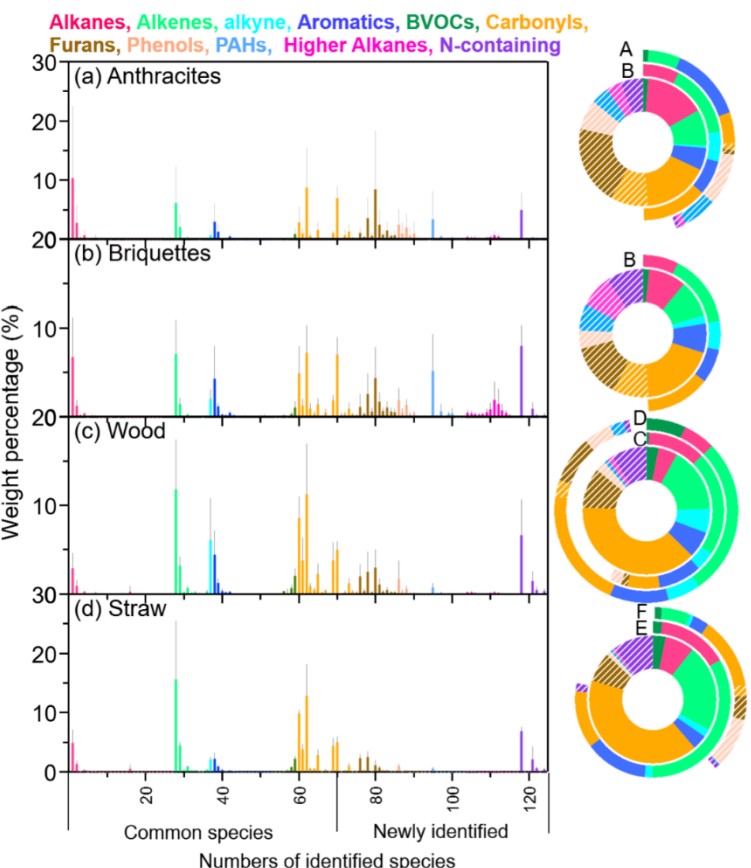

**Figure 3.** Source profiles and compositions of the four types of residential fuels including (a) anthracites, (b) briquettes, (c) wood and (d) straws (average of corncob, bean straws and corn straws) as well as those of the previous results. A. anthracites combustion in stove from Cai et al., 2019 (Cai et al., 2019); B. residential coal combustion simulation in lab from Mo et al., 2016 (Mo et al., 2016); C. hardwood combustion simulation in lab from Stockwell et al., 2016 (Stockwell et al., 2016); D. Black Spruce combustion simulation from lab experiments reported by Hatch et al., 2017 (Hatch et al., 2017); E. wood

combustion simulation from Wang et al., 2014 (Wang et al., 2014); F. rice straw combustion simulation from Stockwell et al., 2015 (Stockwell et al., 2015). The identified species corresponding to the numbers of x axis are listed in Table S2. The shaded
parts of pie charts represent the newly identified species.

## 3.2 SOAP and OHR underestimation from newly identified species

To further understanding the role of residential combustion emitted ROGs in atmospheric chemistry, the hydroxyl radical reactivity (OHR) and secondary organic aerosol formation potential (SOAP) of per unit mass (or concentration) of ROGs emitted from residential combustion was calculated based on the source profiles. OHR is defined here as the sum of hydroxyl
radical (OH) reactivity of each species, calculated by product of ROG species weight percentage ($W_{ROGi}$) in emissions from residential combustion and corresponding OH reaction rate ($k_{OH+ROGi}$) (Carter, 2008; Koss et al., 2018), as presented in Eq. (2) below. SOAP is the sum of SOAP of each species and calculated by multiplying proportion of ROG species with respective SOA yields, as presented in Eq. (3) below.

$$OHR = \sum W_{ROGi} \times k_{OH+ROGi} \tag{2}$$

$$SOAP = \sum W_{ROGi} \times Yield_{ROGi} \tag{3}$$

Among 80 SOA potential precursors in Table S3, the SOA yields of 44 species from previous chamber studies have been published, while SOA yields of nearly half potential precursors were still unknown. The SOA yields in real atmosphere are dependent on nitrogen oxides ($NO_x$) level, total organic aerosol (OA) mass loading and temperature, etc., by modulating the chemical reaction pathway and phase partitioning. The SOA yields mainly measured under high-$NO_x$ conditions ($[NO_x] > 1$
ppb) except for benzenediols ($C_6H_6O_2$) and C2 phenols ($C_8H_{10}O$) from previous chamber studies were scaled to the ambient conditions ($[OA] = 15.0 \ \mu g \ m^{-3}$, T = 25 °C) (Gao et al., 2019) based on the two-product model (Ng et al., 2007; Li et al., 2016) and further corrected for vapor wall losses (Zhang et al., 2014). Table S3 summarized the corrected SOA yields applied in this study and some details of chamber experiments (eg. the chamber yields and the numbers of experiments). Potential precursors with unknown SOA yields include furans, phenols, 3-ring PAHs, terpenes except for α-pinene and alkanes with more than 6
carbon atoms especially branched alkanes. Alkanes containing 13, 14, 16 atoms were estimated using the reported two-product parameters (Presto et al., 2010) which derived from the experimental yields of C12 alkanes and C17 alkanes. SOA yields for other potential precursors were assumed as the corrected SOA yield of species with similar structure or the same number of carbon atoms applied in this study. The overall uncertainty of estimated SOAP is related to the uncertainty of SOA yields and species proportions in source profiles. The yield uncertainty for corrected SOA yields from publications with at least 2
experiments was estimated to be within 11% according to the bias between two-product model fitting results and experimental yields. For other species with published SOA yield from only a single experiment or assumed SOA yields, the yield uncertainty has been estimated as ~50% (Bruns et al., 2016), which was cited in this study. The uncertainty of species proportion was 20% and 34% in coal and biomass combustion profiles, respectively, as mentioned in Section 2.3.4. Thus, the total uncertainty of SOAP could be calculated using error propagation function, being 32% and 41% for coal and biomass combustion, respectively.

Figure 4 shows the OHR and SOAP of per unit mass (or concentration) ROG emissions. The OHR for coal and biomass emissions were quite similar (0.14-0.16 $s^{-1}$ $\mu g^{-1}$ $m^3$) but with different compositions, which was expected as the differences of their ROG compositions. The OHR was dominated by oxygenated species (39.6%-73.7%) and alkenes (15.0%-48.2%), and the contribution of other species was within the range of 9.3%-17.1%. The newly identified ROG species dominated the OHR of coal combustion with the fractions of 64.2% ± 7.8% and 54.6% ± 9.3% for anthracites and briquettes combustion, respectively, due to the large contribution of furans, phenols, PAHs, and higher alkanes in ROGs. In comparison, the previously reported species contributed more to OHR of biomass combustion than that of newly identified species. The ratio of OHR between newly identified and previously reported ROGs was 1.20-1.80 for coals and 0.22-0.51 for biomass, much higher than the ratios of their emissions (0.79-0.81 for coals and 0.20-0.36 for biomass). It meant that the OHR of ROG emissions from residential coal and biomass combustion was underestimated by 59.4% ± 4.8% and 26.2 ± 6.8%, respectively, without the

newly identified species.

SOAP derived from per unit mass ROG emissions of coal combustion was 0.078-0.085 $\mu g$ $\mu g^{-1}$, much higher than that from biomass combustion (0.016-0.035 $\mu g$ $\mu g^{-1}$). Nevertheless, for all samples, newly identified ROGs accounted for over 70% of the SOAP. SOAP was dominated by newly identified oxygenated species like phenols which contributed 47.6% ± 12.4% and 56.7% ± 7.0% to SOAP of emissions from coal and biomass combustion, respectively, and higher alkanes and PAHs also

played important roles in SOAP emissions from coal combustion. The ratios of SOAP derived from newly identified ROGs and previously reported ROGs were 7.8-8.8 and 2.2-2.7 for coals and biomass, respectively, much higher than those of mass percentages. These results indicated that for both coal and biomass combustion, the measurement of newly identified ROGs would be greatly affected on SOA estimation. The field study has found that newly identified ROGs like higher alkanes and PAHs contributed more than 60% of SOA formation from measured precursors in ambient air (Wang et al., 2020a). Our study

of ROG emissions could donate to the explanation of the high SOA formation in atmosphere to some extent. In other words, failure to include newly identified ROGs in emission inventories and SOA models could lead to significant underestimation of residential contribution to SOA production.

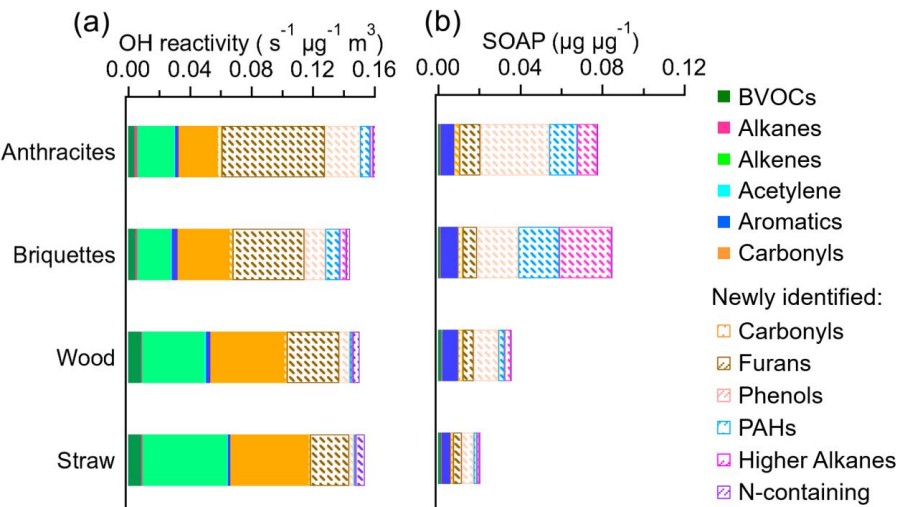

**Figure 4.** The hydroxyl radical (OH) reactivity and SOA formation potential (SOAP) of per unit mass (or concentration) of ROG emissions for coal and biomass combustion. The shaded parts of bar charts represent the newly identified species.

### 3.3 ROG emissions from residential combustion in China

ROG emissions could generally be calculated through multiplying the activity data by the emission factors which were not measured in this study. However, the emission factors of most of the common ROGs like hydrocarbons in the emissions from residential combustion have been reported in previous studies (Cai et al., 2019; Stockwell et al., 2015). Hence, the emission factor of the newly identified ROG species in this study could be estimated by the reported emission factor (EF) of the previously reported species combining with their ERs from residential combustion obtained in this study. Here, benzene as well as its reported EF was used for the purpose above, as benzene was the major overlapping species in all the studies with the high abundance and a relative low uncertainty in combustion emission. The major studies of the EF of benzene from residential combustion were further reviewed and listed in Table S4. Considering the coal samples tested in this study were anthracite and briquette coal, the present study cited the latest reported EF of benzene from anthracite coal combustion by Cai et al. (2019), which agreed with the other reported values of anthracite / briquette coal combustion (Tsai et al., 2003) and 1-2 magnitude lower than those of bituminous coal combustion (Liu et al., 2017a; Liu et al., 2015; Cai et al., 2019; Tsai et al., 2003; Wang et al., 2013). In terms of straw combustion, the present study used the median value of the reported EF of benzene from straw combustion in China, being of 284 mg/kg, which was derived from the simulated real-world combustion in the FLAM-4 laboratory campaign (Stockwell et al., 2015). More particular consideration about selection of the reported EF of benzene was described in the Supplementary Information. We also tested other species with reported EFs (Fig. S12). There were no significant differences (-39%-4% for straws and 6%-26% for coals) of the estimated EFs of newly identified ROGs among different tests, which further confirmed our results were comparable with the previous studies but with more ROG species measured, as shown in Fig. S12. To relate the ROG EFs especially previously unmeasured or rarely measured species

emissions to benzene EF, the ER of ROG species to benzene was the ratio of their concentrations in the sample, and the average ER in different samples of each type of fuel was used in this study, as listed in Table S5. The key point of the relating above was assuming the ERs obtained in this study were consistent with those of the previous studies. The consistence could be confirmed by the good correlation (R=0.73 for straw combustion, and R=0.82 for coal combustion) of the ERs of the overlapping species between our study and the previous studies, as shown in the Fig. S6 (a) and (e) in the Supplementary

Information presented the correlation.

    The estimated EFs of anthracite and straw combustion were used below to estimate the ROG emissions of residential coal and straw combustion in mainland China. Notably, the appliable data about the contribution of bituminous and anthracite coal were from the rural energy survey conducted about ten years ago (2013-2014), which indicated the bituminous coal contributed 97% and 55% of the residential coal consumptions in Baoding (Zhi et al., 2017; Zhi et al., 2015) and Beijing (Zhao et al., 2015).

China has been carrying out toughest-ever clean energy substitution and vigorously replacing bituminous coal with anthracite in response to the cleaning action plan in the residential sector since 2013, which were further strengthened from 2017 to 2020 during the three-year battle against air pollution (eg. Action Plan for Clean and Efficient Utilization of Coal (2015-2020), http://zfxxgk.nea.gov.cn/). The use and sale of bituminous coal were generally not allowed (Luo, 2019). Thus, we could expect the large decrease of the use of bituminous coal in residential sector in China although there is no updated statistical data

appliable. This study assumed that anthracite is the main residential coal type to roughly estimate ROG emissions in China. Accordingly, the national ROG emissions of residential combustion were estimated combining with the residential coal consumption and the crop straw combustion data in China. Specifically, the data of residential coal consumption from 2010 to 2019 were from the China Energy Statistical Yearbook (National Bureau of Statistics, 2010-2022), which included the data of each province in China mainland in each year. The data of crop straw combustion from 2010 to 2019 (Table S6) were from

Report of Prospects and Investment Strategy Planning Analysis on China Straw Refuse Treatment Industry (2022-2027) (Qianzhan Industrial Research Institute, 2022), which only reported the total amount of the whole China mainland and included both the household and field combustion. The province data of crop straw combustion in 2017 (Table S7) were used to study the spatial distribution in this study, which were from Second National Pollution Source Census Bulletin (Ministry of Ecology and Environment of the People's Republic of China et al., 2020).

The spatial and temporal distribution of ROG emissions from residential combustion was presented in Fig. 5. The total emissions of ROGs from residential coal combustion and crop straw combustion were 14 kt and 4384 kt in 2019, respectively, and as expected these values were underestimated by 44.3% ± 11.8% and 22.7% ± 3.9%, respectively, because fewer species were included previously. Unexpectedly, the ROG emissions from crop straw combustion were two orders of magnitude higher than those of coal combustion, which included those both from household and field combustion. Even if the emission factors

of bituminous coal were applied, the ROG emissions from residential coal combustion in China would increase by approximately 1-2 orders of magnitude, which were still lower than the ROG emissions from biomass combustion. More refined energy consumption statistics are necessary to update as adjustment of Chinese energy structure. Notably, the straw combustion emissions were stable after 2017 compared with the gradual decrease from 2010 to 2017, mainly due to the

limitation of straw utilization (Zhu et al., 2019). In comparison, ROG emissions of residential coal combustion began to decrease after 2017, benefiting from the clean heating action in the north of China (National Development and Reform Commission, 2017). Spatially, the hot areas of ROG emissions from residential coal combustion were mainly in the North China Plain (NCP) (Yang et al., 2020), while those of straw combustion were mainly in the main food-production bases of China like the northeastern China, the southern NCP, and Jiangsu, Anhui, and Hunan provinces.

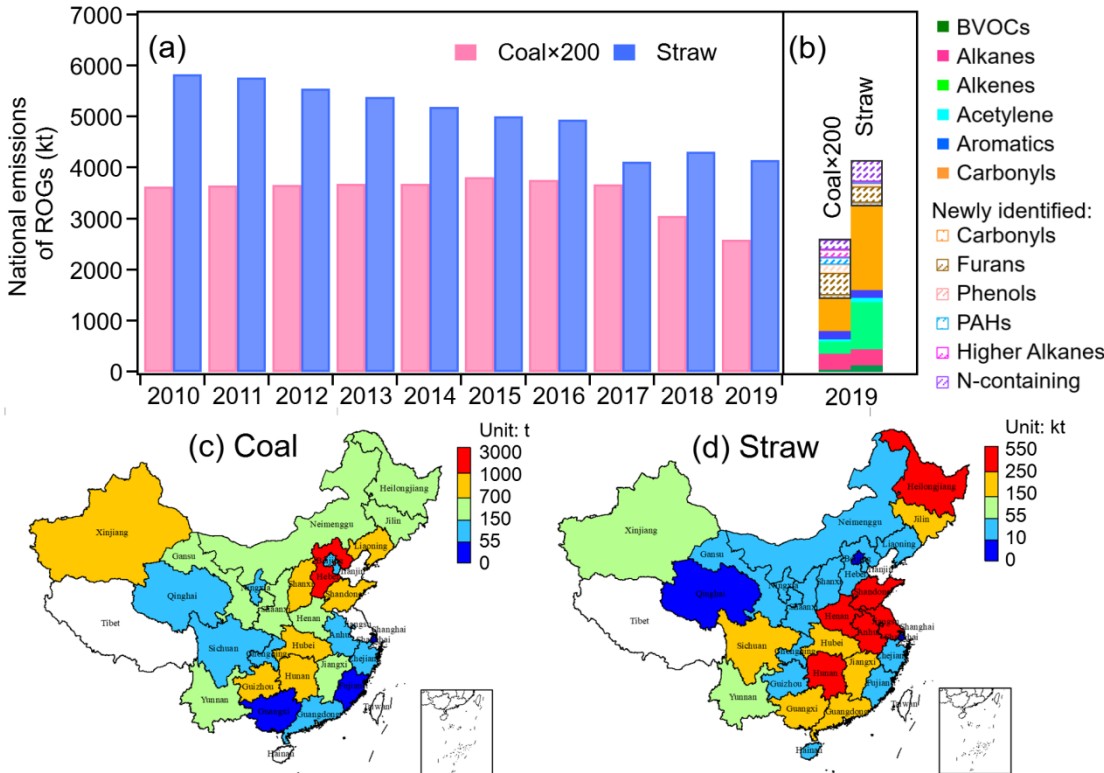

**Figure 5.** Emissions of ROGs from residential coal and straw combustion in China. (a) Annual variation of national ROG emissions and (b) the speciation of ROG emissions in 2019 from residential coal and straw combustion. The black boxes indicate the emissions of newly identified species. (c, d) Spatial distribution of ROG emissions from coal combustion in 2019 and straw combustion in 2017 respectively. The blank parts on the map indicate the provinces with missing data.

## 4. Conclusions

In this study, a near-complete chemical description of ROGs emitted from residential combustion in rural household of China was developed through quantifying all signals by $H_3O^+$ PTR-ToF-MS and supplementing C2-C22 aliphatic hydrocarbons by GC-MS/FID and $NO^+$ PTR-ToF-MS. Among the near-complete description of ROGs, 55 species un- and under-characterized in previous studies using GC methods were analyzed intensively by $H_3O^+/NO^+$ PTR-ToF-MS, mainly including oxygenated species (carbonyls, furans, phenols), higher hydrocarbons (PAHs, higher alkanes) with carbon atoms more than 8 as well as

nitrogen-containing compounds. Compared to the comprehensive measurements by more instruments previously, the combination of PTR-ToF-MS and the GC-MS/FID method was labor-saving and further could minimize the measurement uncertainties from the synthesis of measurement data due to fewer kinds of instruments.

For the nearly complete ROGs dividing into three categories by the element composition, oxygenated species played a similar major role with hydrocarbons, and nitrogen-containing species dominated by acetonitrile and acrylonitrile were also

considerable in ROG emissions from residential combustion. Especially, coal combustion emitted considerable higher alkanes including 8-21 carbon atoms, gaseous PAHs (mainly including 2-3 benzene rings), furans and phenols, differently biomass combustion emitted more carbonyls and terpenes.

Considering the newly discovered species, it is observed that approximately half and a quarter of the ROG emissions from coal and biomass combustion are underestimated. Combining with the spatial-temporal consumption of residential coal and

biomass combustion in China, ROG emissions of residential combustion were estimated. The ROG emissions from straw combustion were two orders of magnitude higher than those from coal combustion with negligible decline in recent years as the limited straw utilization ratio, which suggested the biomass combustion would be the only important residential emissions with the continuous replacement of residential coal in rural China. Given the newly identified species more reactive or with higher SOA yields, amplified underestimation of OHR and SOAP was observed for both coal combustion ($59.4\% \pm 4.8\%$ and

$89.2\% \pm 1.0\%$) and biomass combustion ($26.2\% \pm 6.8\%$ and $70.3\% \pm 1.6\%$). These results highlighted the importance of the completely speciated measurement of the ROG emissions from residential combustion.

**Data availability**

Data presented in this paper are freely accessible from the following link: https://data.mendeley.com/datasets/z78zz7mv7h/2 (Mendeley Data, V2, doi: 10.17632/z78zz7mv7h.2, Wang et al., 2022)

**Competing interests**

The authors declare that they have no conflict of interest.

**Authors contributions.** All authors contributed to the manuscript and have given approval of the final version. H.W. and C.H. designed the study. H.W. and Y.G. performed the data analyses and wrote the manuscript. Y.G., L.Y., and S.J. conducted the

experiment. B.Y., G.S., Y.L., Q.W., D.H., S.Z., and S.L., contributed to the interpretation of results. L.Z and A.K. revised the manuscript. S.T. assisted in sampling.

**Acknowledgments.** This work was supported by National Natural Science Foundation of China (42175135), the National Key R&D Program of China (2022YFE0136200), and the Science and Technology Commission of the Shanghai Municipality (20ZR1447800).

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
