# Peer review of "Measurement report: Underestimated reactive organic gases from residential combustion: insights from a near-complete speciation"

_Atmospheric Chemistry and Physics, 2022_

## Author Response (AR1)

**Responses to the Reviewer's Comment (RC#1)**

We thank the reviewer for the constructive and insightful comment. Our point-by-point responses can be found below, with reviewer comments in **black**, our responses in **blue**, alongside the relevant revisions to the manuscript in **red**.

**General comments:**

This study constructs a near-complete chemical description of ROGs emitted from residential coal and biomass combustion using GC-MS/FID and Vocus PTR-ToF-MS. ROG emissions were underestimated by 44.3 ± 11.8% and 22.7 ± 3.9%, considering the un-characterized species, which resulted in the underestimation of SOA formation potential and OH reactivity. In the current framework, 125 species were quantified, accounting for approximate 90% of the total ROG emission. SOAP and OHR were re-estimated incorporating the newly identified species and the spatial distribution and annual variation of ROG emissions from residential coal and straw combustion in China are reported. The topic is covered by ACP and the data is of interests to relevant readers. However, novelty is lacking with respect to methodology and results and the discussion lacks depth. Also, the manuscript is not well drafted, mistakes and errors are commonly seen.

**R:**

[revised manuscript text omitted]

We have revised it in the sections of Abstract and Introduction in the revised manuscript.

**Table R1** Review on the measurements of ROG emissions from the residential combustion.

| Method | Fuel types | Reference | Numbers of species | | |
|---|---|---|---|---|---|
| | | | GC-MS/FID | PTR-ToF-MS | Other instruments |
| Combustion in stove | Residential coals, biomass (straw and wood) from China | This study | 71 | 84 (84-92% of the overall peak mass) | — |
| | Corncob from China | (Wu et al., 2022) | — | 13 | — |
| | Anthracite, bituminous coal from China | (Cai et al., 2019) | — | 79-89 (90-96% of the overall peak intensities) | — |
| | Beech (Fagus sylvatica) logs | (Bruns et al., 2017) | — | 64 (94-97% of the total mass) | — |
| | Biomass (peanut shell, maize straw), raw coals from China | (Wang et al., 2013) | 60 | — | 24 (DNPH-HPLC [a]) |
| Combustion simulation | Biomass fuels from the western US | (Akherati et al., 2020) | — | 150 | — |

| Method | Fuel types | Reference | Numbers of species | | |
| | | | GC-MS/FID | PTR-ToF-MS | Other instruments |
|---|---|---|---|---|---|
| in Lab | Burned fuels from the western US | (Koss et al., 2018) | — | 172 (~95% of the overall peak intensities) | 15 (FTIR [b]), 261(GC-CIMS [c]) |
| | Four burns: ponderosa pine boughs, Chinese rice straw, Indonesian peat, and black spruce boughs | (Hatch et al., 2017) | ~27 | ~71 | ~13 (FTIR) ~418 (GC×GC) |
| | Authentic globally significant fuels | (Stockwell et al., 2015) | — | 46-92 | — |
| | Biomass burns of 18 fuel types from 3 geographic regions in the US | (Gilman et al., 2015) | 187 | Unpublished | Unpublished (FTIR) |
| | Residential coal, rice, maize, and wheat straw from China | (Mo et al., 2016) | 62 | — | 13 (DNPH-HPLC) |
| | biomass, residential coal from China | (Liu et al., 2008) | 92 | — | — |

Note:

[a], DNPH-HPLC: 2, 4-Dinitrophenyl hydrazine followed by high performance liquid chromatography (HPLC).

[b], FTIR: Fourier transform infrared spectroscopy.

[c], GC-CIMS: Gas chromatography chemical ionization mass spectrometry.

**Specific comments:**

**Q1:** Line 113-116. PTR-MS with $H_3O^+$ detects the VOC molecules with higher proton affinity than water and PTR-MS with $NO^+$ extents the range of compounds that can be detected. As you mentioned, a total of 1005 peaks were extracted with m/z lower than 200 in $H_3O^+$ mode and selected peaks in $NO^+$ mode.

(1) What are the range of compounds that $NO^+$ mode supplements and the criteria of selecting these compound species?

**R:**

$NO^+$ PTR-ToF-MS in this study is only used for a supplementary measurement of higher alkanes (≥C8). The criteria of selecting these species were that (1) the species could not be measured well by $H_3O^+$ PTR-ToF-MS, (2) the methods of which have been well-established by $NO^+$ PTR-ToF-MS previously. More details are as follows.

Multi-instruments were deployed to measure ROGs in the emissions of residential combustion, aiming to develop a near-complete speciation of ROGs in this study, which included $H_3O^+$ PTR-ToF-MS, $NO^+$ PTR-ToF-MS, and GC-MS/FID. $H_3O^+$ PTR-ToF-MS could identify the ROGs as long as the proton affinity of ROGs greater than that of water (691 kJ mol$^{-1}$) (Yuan et al., 2017),

with relatively complete species coverage. It may be the preferred method toward ROGs complete measurement, because most ROGs can be detected (Li et al., 2020; Krechmer et al., 2018) and the sensitivity for a given ROG can be calculated theoretically (Sekimoto et al., 2017) by $H_3O^+$ PTR-ToF-MS.

Despite these advantages, $H_3O^+$ PTR-ToF-MS has two limitations related to the reagent ion chemistry. Firstly, the technique is insensitive to C2-C7 alkanes, ethene and acetylene with lower proton affinity than water (Jobson et al., 2009). Here, these common constituents of urban atmospheres were well-measured by traditional technology such as GC-MS/FID. Secondly, higher alkanes (≥C8), one kinds of important components of the fuel combustion (Huo et al., 2021; Jathar et al., 2014), were difficult to quantify by $H_3O^+$ PTR-ToF-MS due to fragments produced during the ionization process. $NO^+$ PTR-ToF-MS has been demonstrated to provide a supplementary measurement of higher alkanes (Wang et al., 2020a; Koss et al., 2016).

Different from $H_3O^+$ PTR-ToF-MS by which the sensitivity for a given ROG can be calculated theoretically even without the standard for calibration, authentic standards are necessary for quantification by $NO^+$ PTR-ToF-MS, which limited the characterization of mass spectrum ionized by $NO^+$. The difficulty to predict the ionized ROG products and to interpret the mass spectrum unambiguously further limited its application in ROG speciation, because $NO^+$ has three common reaction mechanisms with ROGs: charge transfer, hydride abstraction, and cluster formation (Koss et al., 2016). Therefore, $NO^+$ PTR-ToF-MS in this study was only used for a supplementary measurement of higher alkanes (≥C8) with a well-established quantitative method (Wang et al., 2020a).

(2) What are the organic gases that are not sensitively detected by PTR-MS with $H_3O^+$ and $NO^+$ modes and how can GC/MS-FID complement the measurement? This information should be included, either in the main text or the SI. Currently, the analysis section is brief.

**R:**

The two limitations of PTR-ToF-MS with $H_3O^+$ on organic gas detection are given in the reply to the previous question Q1(1). Briefly, the technique is limited in the detection of aliphatic hydrocarbons involving alkanes, alkenes and alkynes. $NO^+$ PTR-ToF-MS has been demonstrated to provide a supplementary measurement of higher alkanes (≥C8) (Wang et al., 2020a; Koss et al., 2016). With $NO^+$ PTR-ToF-MS, light alkanes and unsaturated aliphatic hydrocarbons were not yet well measured, which were subject to the interference from fragments producing by many different species. The GC-MS/FID could provide a supplementary measurement of these species against PTR limitations. Besides, GC/MS-FID provided isomer speciation which is incapable for PTR-ToF-MS with both $H_3O^+$ and $NO^+$.

We have clarified it more clearly in Section 2.2 and Section 2.3 of the revised manuscript as following:

Section 2.2.4, the first paragraph:

"$H_3O^+$ PTR-ToF-MS could identify the ROGs as long as the proton affinity of ROGs greater than that of water (691 kJ mol$^{-1}$) (Yuan et al., 2017), with relatively complete species coverage. It may be the preferred method toward ROG complete measurement, because most ROGs can be detected

(Li et al., 2020; Krechmer et al., 2018) and the sensitivity for a given ROG can be calculated theoretically by $H_3O^+$ PTR-ToF-MS (Sekimoto et al., 2017). Despite these advantages, $H_3O^+$ PTR-ToF-MS has two limitations related to the reagent ion chemistry. Firstly, the technique is insensitive to C2-C7 alkanes, ethene and acetylene with lower proton affinity than water (Jobson et al., 2009). Secondly, higher alkanes ($\geq$C8), one kinds of important components of the fuel combustion (Huo et al., 2021; Jathar et al., 2014), were difficult to quantify by $H_3O^+$ PTR-ToF-MS due to fragments produced during the ionization process."

Section 2.3.1, the first paragraph:

"$NO^+$ PTR-ToF-MS has been demonstrated to provide a supplementary measurement of higher alkanes (Wang et al., 2020a; Koss et al., 2016). Different from $H_3O^+$ PTR-ToF-MS by which the sensitivity for a given ROG can be calculated theoretically even without the standard for calibration, authentic standards are necessary for quantification by $NO^+$ PTR-ToF-MS, which limited the characterization of mass spectrum ionized by $NO^+$. The difficulty to predict the ionized ROG products and to interpret the mass spectrum unambiguously further limited its application in ROG speciation, because $NO^+$ has three common reaction mechanisms with ROGs: charge transfer, hydride abstraction, and cluster formation (Koss et al., 2016). Therefore, $NO^+$ PTR-ToF-MS in this study was only used for a supplementary measurement of higher alkanes ($\geq$C8) with a well-established quantitative method (Wang et al., 2020a)."

Section 2.3.2, line 215-217:

"Against PTR limitations (Arnold et al., 1998; Gueneron et al., 2015), GC-MS/FID with a cryogen-free preconcentration device was also deployed."

**Q2:** Figure 1, when conducting the mass closure analysis, how to quantify the organic gases that are not effectively detected by PTR-MS and GC-FID/MS. Given FID is adopted, THC data (if any) may help.

**R:**

In this study, we assumed the "total" ROGs including all signals quantified by $H_3O^+$ PTR-ToF-MS and supplementing C2-C22 aliphatic hydrocarbons measured by GC-MS/FID and $NO^+$ PTR-ToF-MS, mainly due to the broad species coverage of PTR. Koss et al. (2018) made an intensive comparison of different instruments for ROG measurements of emissions of biomass combustion, including GC pre-separation, two-dimensional GC system (GC×GC), fourier transform infrared spectroscopy (FTIR), $NO^+$ chemical ionization mass spectrometer ($NO^+$ CIMS) and $H_3O^+$ PTR-ToF-MS. The comparison demonstrated that $H_3O^+$ PTR-ToF-MS might be the most suitable for the detection of the lowest-volatility and most polar species, which covered the most (50%-79%) species, comparing with the other instruments, in the combined ROG measurement covering more than 500 species from different instruments (Koss et al., 2018). Thus, $H_3O^+$ PTR-ToF-MS might be a preferent and promising method for the development of near-complete ROG speciation relevant for residential combustion, but need to combine the measurement of aliphatic hydrocarbons. Here, the C2-C22 aliphatic hydrocarbons, not well measured by $H_3O^+$ PTR-ToF-MS, have been supplemented by GC-MS/FID and $NO^+$ PTR-ToF-MS. From the technological point of view, the

ROGs by simplified instrument deployment in this study might represent the total ROGs to some extent based on all current state-of-the-art technologies.

As the reviewer suggested, the total hydrocarbons (THCs) provide a potential path towards mass closure (carbon budget). The feasibility for mass closure analysis using the THCs data has been discussed in terms of the instrument measurement fundamentals and the experimental results as below.

(1) Measurement fundamentals

The concentration of total hydrocarbon in gas samples is generally detected by the flame ionization detector (FID). According to the FID ionization mechanism, the main reason of signal detection by FID is that the carbon atoms of organic compounds produce $CHO^+$ in the hydrogen flame. Thus, the number of carbon atoms in organic compounds per unit mass determine the FID response value of organic compounds (Sun, 1979). However, the types of organic compounds affect the efficiency of $CHO^+$ formation. For most hydrocarbons, such as normal and isomeric alkanes, alkenes etc., the response signal per gram of carbon should be relatively consistent. While, FID detector has a more variable and smaller response to oxygenated organic compounds than hydrocarbons. Only some carbon could form positive ions when carbon and oxygen are connected by single bond, and it is almost impossible for carbon to form positive ions when double bonds occur between carbon and oxygen (Fan et al., 2002).

The photoionization detector (PID) is an alternative technology for THCs measurement. Similar to the FID, the PID also converts the organic compounds into an ion stream for detection, while the energy of ionization is provided by an ultraviolet lamp instead of a hydrogen flame. The PID techniques also suffers from low and variable response towards oxygenated organic compounds (Yang and Fleming, 2019; Cai et al., 2019).

(2) Experimental results

Cai et al. (2019) has made an attempt on comparison between the THCs (except methane) detected by a portable analyzer (JiShunAn JK40, Shenzhen, China) equipped with a PID module (noted as PID) and the sum of organic species (ΣROG) detected by the PTR-ToF-MS. THC emissions by PID were always lower than ΣROG emissions of the PTR-ToF-MS throughout the combustion cycle (Cai et al., 2019). Thus, it is expected that the ROG emissions detected by PTR-ToF-MS and GC-MS/FID in current work exceed THC emissions and mass closure analysis using THCs is not appropriate.

Therefore, this work overlooked the organic gases not effectively detected by PTR-MS and GC-MS/FID and highlighted the detailed chemical understanding based on all current state-of-the-art technologies. The total ROGs requires further advances in directly analytical measurements.

**Q3:** Line 118, 162 ions with a relative high degree of certainty… Again, the introduction of the data treatment is inadequate. What is the standard of selecting these ions and how to determine the certainty threshold. Similar question goes to line 156, 87 out of 162 species were used….please give more details regarding data treatment.

**R:**

In order to clearly introduce the data treatment of deployed instruments in this study, $H_3O^+$ PTR-ToF-MS is introduced separately from $
[revised manuscript text omitted]

We have revised the description of measurement methods in detail in Section 2.2 and Section 2.3 in the revised manuscript.

**Q4:** Line 122…. assuming all the signals with the same sensitivity as acetone. Does this assumption stand, since acetone has a higher proton affinity than many VOC compounds.

**R:**

It should be clarified that the sensitivity of acetone was employed as a proxy only for the 861 unknown masses measured by $H_3O^+$ PTR-ToF-MS in this study. Acetone has been the preferred choice for sensitivity substitutes of other species (Cai et al., 2019) due to its humidity independence (De Gouw and Warneke, 2007). Acetone was also used in this study to keep consistent with previous studies. As the reviewer mentioned, the proton affinity and the kinetic reaction rate constant ($k_{kinetic}$) of acetone are 812 kJ $mol^{-1}$ and $3.23 \times 10^{-9}$ $cm^3$ $molecule^{-1}$ $s^{-1}$ respectively, higher than those of ~80% of species in Cappellin's study on $k_{kinetic}$ (Cappellin et al., 2012). Thus, the upper limit to the sensitivity was generally provided for unknown species (Koss et al., 2018). For unknown masses, the uncertainty was mainly from the difference of their "real" sensitivities from the assumed sensitivity of acetone, which was dependent on the difference of their $k_{kinetic}$. Generally, the oxygenated species ($C_xH_yO_z$) have higher (~2.5-3.5×$10^{-9}$ $cm^3$ $mol^{-1}$ $s^{-1}$) $k_{kinetic}$ than those of aliphatic hydrocarbons ($C_xH_y$ , ~2×$10^{-9}$ $cm^3$ $mol^{-1}$ $s^{-1}$), and the $k_{kinetic}$ of acetone is 3.23×$10^{-9}$ $cm^3$ $mol^{-1}$ $s^{-1}$ (Cappellin et al., 2012). Thus, the maximum uncertainty of the estimated concentrations of unknown masses caused by this assumption is estimated to be 65% although the "real" $k_{kinetic}$ is as low as 2×$10^{-9}$ $cm^3$ $mol^{-1}$ $s^{-1}$ in this study.

The reason for using the sensitivity of acetone and the potential uncertainty has been added in Section 2.2.3 of the main text.

"**Estimation method (861 masses).** For unknown masses, it is difficult to estimate $k_{kinetic}$ which depends on molecular properties, and the sensitivity of acetone was used as a proxy (Cai et al., 2019).

…For unknown masses, the uncertainty was mainly from the difference of their "real" sensitivities from the assumed sensitivity of acetone, which was dependent on the difference of their $k_{kinetic}$. Generally, the oxygenated species ($C_xH_yO_z$) have higher (~2.5-3.5×$10^{-9}$ $cm^3$ $mol^{-1}$ $s^{-1}$) $k_{kinetic}$ than those of aliphatic hydrocarbons ($C_xH_y$ , ~2×$10^{-9}$ $cm^3$ $mol^{-1}$ $s^{-1}$), and the $k_{kinetic}$ of acetone is 3.23×$10^{-9}$ $cm^3$ $mol^{-1}$ $s^{-1}$ (Cappellin et al., 2012). Thus, the maximum uncertainty of the estimated concentrations of unknown masses caused by this assumption is estimated to be 65% although the "real" $k_{kinetic}$ is as low as 2×$10^{-9}$ $cm^3$ $mol^{-1}$ $s^{-1}$ in this study."

**Q5:** Line 169 why is formaldehyde not reported in your study since PTR ionize formaldehyde efficiently.

**R:**

Formaldehyde, a special species during PTR measurement was not detected in this study, suffering from double effects by (1) mass discrimination and (2) reversible reaction.

(1) Mass discrimination

For the PTR-ToF-MS deployed in this study, as the transmission curve (Fig. S2) shows, the mass transmission efficiency of protonated formaldehyde at m/z 31 is close to 0, leading to an almost negligible sensitivity of formaldehyde. The decisive factor for transmission efficiency of PTR-ToF-MS is the radio frequency (RF) amplitude and the frequency of the big segmented quadrupole (BSQ), which can be set up as an ion filter (Wang et al., 2020c). A similar situation has been reported by Yang et al. (2022), in which formaldehyde was not measured by the PTR-ToF-MS (Vocus 2R, Aerodyne Research Inc.) (Yang et al., 2022).

(2) Reversible reaction

Non-negligible back reactions between protonated formaldehyde and water vapor can reduce the sensitivity for formaldehyde (Spanel and Smith, 2008; Cui et al., 2016). For the developed Vocus instrument deployed in current work, the water vapor flow increases from the previous level 4−8 sccm ($cm^3$ $min^{-1}$ at 105 Pa and 273.15 K) (De Gouw et al., 2004) to 20−30 sccm (Krechmer et al., 2018). This change brings a great advantage that the humidity dependence of instrument response to most species can be ignored. However, for formaldehyde, the reduction of its sensitivity is predictable because the really high water vapor concentration was provided for the back reactions.

We have clarified it more clearly in Section 2.2.4 in the revised manuscript.

**Q6:** Line 255 figure 2. What is the x-axis label, m/z or numbers of identified species. Add x-axis label.

**R:**

The x-axis label "Numbers of identified species" was added at the bottom of Fig. R4 (Fig. 3 in the revised main text). An additional explanation of x axis was provided in the title of Fig. R4, that is "The identified species corresponding to the numbers of x axis are listed in Table S2 in the Supplementary Information".

[Figure]

**Figure R4.** Source profiles and compositions of the four types of residential fuels including (a) anthracites, (b) briquettes, (c) wood and (d) straws (average of corncob, bean straws and corn straws) as well as those of the previous results. A. anthracites combustion in stove from Cai et al., 2019 (Cai et al., 2019); B. residential coal combustion simulation in lab from Mo et al., 2016 (Mo et al., 2016); C. hardwood combustion simulation in lab from Stockwell et al., 2016 (Stockwell et al., 2016); D. Black Spruce combustion simulation from lab experiments reported by Hatch et al., 2017 (Hatch et al., 2017); E. wood combustion simulation from Wang et al., 2014 (Wang et al., 2014); F. rice straw combustion simulation from Stockwell et al., 2015 (Stockwell et al., 2015). The identified species corresponding to the numbers of x axis are listed in Table S2 in the Supplementary Information. The shaded parts of pie charts represent the newly identified species.

**Technical corrections:**

Line 73, 276, 320, 360 space missing before the left bracket.

**R:**

Space was added before the left bracket as suggested.

Line 94 "particles was" should be "particles were"

**R:**

Revised as suggested.

Line 97 space missing beforeâ„ƒ.

**R:**

Space was added before "°C" as suggested.

Line 100 there were 23 samples were collected, grammatical mistake

**R:**

This sentence has been revised to the following statement:

"Here totally 23 samples were collected, as shown in Fig. S1".

Line 130 space missing before the left bracket.

**R:**

Revised as suggested.

Line 135 10 carbonyls are included in Table S2.

**R:**

In the original manuscript, 12 carbonyls were measured by GC-MS/FID. Among them, as the reviewer mentioned, 10 carbonyls were listed in the category "Carbonyls" in Table S2. It should be noted that the other 2 carbonyls including methacrolein and methyl vinyl ketone were classed into Biogenic Volatile Organic Compounds (BVOCs) in Table S2.

To avoid ambiguity, we revised "carbonyls" in the main text into "oxygenated organic compounds":

"Fifty-seven hydrocarbons and 12 oxygenated organic compounds were measured by GC-MS/FID and quantified using standard gases (Linde gas North America LLC, USA)."

Line 137 delete including 68 species.

**R:**

Revised as suggested.

Line 185 add a space before and after and check throughout the whole manuscript.

**R:**

All issues related to space missing were checked and revised throughout the whole manuscript as suggested.

Line 211 as → that

Line 227 as Fig. 2(c) shown → as shown in Fig. 2(c) or as Fig. 2(c) showed/shows.

**R:**

Revised as suggested.

Line 251 253 two types of data format shown, i.e., xxx% ± xxx% or xxx ± xxx%. Check throughout the whole manuscript keep consistency.

**R:**

The data format was checked and unified to one type throughout the whole manuscript.

Line 276, according SOA yields → respective SOA yields.

**R:**

Revised as suggested.

Line 351, due to is a preposition and should not a complete sentence.

**R:**

"due to" was replaced by "because".

Line 354, comparing with → compared with

Line 357, 2017 benefiting → 2017, benefiting

Line 397, in rural of China → in rural China

**R:**

Revised as suggested.

I believe the above technical corrections are not complete, authors should check carefully.

**R:**

The technical details and grammar have been checked and revised carefully throughout the whole manuscript.


**Responses to the Reviewer's Comment (RC#2)**

We thank the reviewer for the constructive and insightful comment. Our point-by-point responses can be found below, with reviewer comments in **black**, our responses in **blue**, alongside the relevant revisions to the manuscript in **red**.

The manuscript by Gao et al. reported the near-complete speciation of reactive organic gases (ROGs) with 125 species identified to evaluate their emission characteristics from residential combustion. The authors used a Gas Chromatography equipped with a Mass Spectrometer and a Flame Ionization Detector (GC-MS/FID) and $H_3O^+/NO^+$ Proton Transfer Reaction Time-of- Flight Mass Spectrometer (Vocus PTR-ToF-MS) to identify 55 previously un- and under-characterized species. Without considering these "newly identified species", the ROG emissions from residential coal and biomass combustion would be underestimated by $44.3\% \pm 11.8\%$ and $22.7\% \pm 3.9\%$, respectively, which further highlighted the potential underestimation of secondary organic aerosols formation potential (SOAP) and OH reactivity (OHR) of ROG emissions. Overall, this study would be a useful addition to better understanding the detailed speciation of ROGs from residential combustion. However, the novelty of this study should be clearly addressed, especially given that some previous studies also applied these advanced instruments and have identified these "newly identified species" (Figure 2).

**R:**

[revised manuscript text omitted]

We have revised it in the sections of Abstract and Introduction in the revised manuscript.

**Table R1** Review on the measurements of ROG emissions from residential combustion.

| Method | Fuel types | Reference | Numbers of species | | |
| | | | GC-MS/FID | PTR-ToF-MS | Other instruments |
|---|---|---|---|---|---|
| Combustion in stove | Residential coals, biomass (straw and wood) from China | This study | 71 | 84 (84-92% of the overall peak mass) | — |
| | Corncob from China | (Wu et al., 2022) | — | 13 | — |
| | Anthracite, bituminous coal from China | (Cai et al., 2019) | — | 79-89 (90-96% of the overall peak intensities) | — |
| | Beech (Fagus sylvatica) logs | (Bruns et al., 2017) | — | 64 (94-97% of the total mass) | — |
| | Biomass (peanut shell, maize straw), raw coals from China | (Wang et al., 2013) | 60 | — | 24 (DNPH-HPLC [a]) |

| Method | Fuel types | Reference | Numbers of species | | |
|---|---|---|---|---|---|
| | | | GC-MS/FID | PTR-ToF-MS | Other instruments |
| Combustion simulation in Lab | Biomass fuels from the western US | (Akherati et al., 2020) | — | 150 | — |
| | Burned fuels from the western US | (Koss et al., 2018) | — | 172 (~95% of the overall peak intensities) | 15 (FTIR [b]), 261(GC-CIMS [c]) |
| | Four burns: ponderosa pine boughs, Chinese rice straw, Indonesian peat, and black spruce boughs | (Hatch et al., 2017) | ~27 | ~71 | ~13 (FTIR) ~418 (GC×GC) |
| | Authentic globally significant fuels | (Stockwell et al., 2015) | — | 46-92 | — |
| | Biomass burns of 18 fuel types from 3 geographic regions in the US | (Gilman et al., 2015) | 187 | Unpublished | Unpublished (FTIR) |
| | Residential coal, rice, maize, and wheat straw from China | (Mo et al., 2016) | 62 | — | 13 (DNPH-HPLC) |
| | biomass, residential coal from China | (Liu et al., 2008) | 92 | — | — |

Note:

[a], DNPH-HPLC: 2, 4-Dinitrophenyl hydrazine followed by high performance liquid chromatography (HPLC).

[b], FTIR: Fourier transform infrared spectroscopy.

[c], GC-CIMS: Gas chromatography chemical ionization mass spectrometry.

Specific issues:

**Q1:** Line 115-116: why only selected peaks (mainly higher alkanes) under $NO^+$ mode PTR measurements were studied?

**R:**

Multi-instruments were deployed to measure ROGs in the emissions of residential combustion, aiming to develop a near-complete speciation of ROGs in this study, which included $H_3O^+$ PTR-ToF-MS, $NO^+$ PTR-ToF-MS, and GC-MS/FID. $H_3O^+$ PTR-ToF-MS could identify the ROGs as long as the proton affinity of ROGs greater than that of water (691 kJ mol$^{-1}$) (Yuan et al., 2017), with relatively complete species coverage. It may be the preferred method toward ROG complete measurement, because most ROGs can be detected (Li et al., 2020; Krechmer et al., 2018) and the sensitivity for a given ROG can be calculated theoretically by $H_3O^+$ PTR-ToF-MS (Sekimoto et al., 2017).

Despite these advantages, $H_3O^+$ PTR-ToF-MS has two limitations related to the reagent ion chemistry. Firstly, the technique is insensitive to C2-C7 alkanes, ethene and acetylene with lower proton affinity than water (Jobson et al., 2009). Here, these common constituents of urban

atmospheres were well-measured by traditional technology such as GC/MS-FID. Secondly, higher alkanes ($\geq$C8), one kinds of important contents of the fuel combustion (Huo et al., 2021; Jathar et al., 2014), were difficult to quantify by $H_3O^+$ PTR-ToF-MS due to fragments produced during the ionization process. $NO^+$ PTR-ToF-MS has been demonstrated to provide a supplementary measurement of higher alkanes (Wang et al., 2020; Koss et al., 2016).

Different from $H_3O^+$ PTR-ToF-MS by which the sensitivity for a given ROG can be calculated theoretically even without the standard for calibration, authentic standards are necessary for quantification by $NO^+$ PTR-ToF-MS, which limited the characterization of mass spectrum ionized by $NO^+$. The difficulty to predict the ionized ROG products and to interpret the mass spectrum further limited its application in ROG speciation, because $NO^+$ has three common reaction mechanisms with ROGs: charge transfer, hydride abstraction, and cluster formation. Therefore, $NO^+$ PTR-ToF-MS in this study was only used for a supplementary measurement of higher alkanes ($\geq$C8) with a reported well-established quantitative method (Wang et al., 2020).

**Q2:** Line 151: the loss of acids and alcohols in the canister is larger, and the author attributed this to their functional groups of -COOH and -OH. Would it be more direct to relate this to the volatility of compounds? Are there any criteria to exclude these compounds from the analysis?

**R:**

The loss of ROGs in the canisters during storage mainly relate to their volatility and polarity. As shown in Fig. R1(a), there was no significant dependence of the loss proportion on the volatility (the effective saturation vapor concentration, $C^*$) of ROGs in current experimental results for 10 days. The species with similar $\log_{10}C^*$ had largely different loss fractions and the loss fraction of the species with increased $\log_{10}C^*$ didn't reduced regularly as expected. The higher polarity of alcohols is a possible reason for their larger loss proportion in Fig. R1(b), as polar species preferentially adsorbed on surface sorption sites of SUMMA canister inner walls (Batterman et al., 1998). Considering the rough rank of functional groups involved in this study from high polarity to low polarity (-COOH, -OH, -CN, $-NO_n$, -CHO, RCO- and single/double/triple bond), species with higher polarity like acids and alcohols might have larger loss proportion, which were excluded from the analysis.

[Figure]

**Figure R1.** Plots of the loss fraction versus volatility (the logarithmic effective saturation vapor concentration, $\log_{10}C^*$) and polarity (functional groups) of species respectively. (a) The scatter plot of the loss fractions of individual species and $\log_{10}C^*$, colored by functional groups (see legend). (b) The average and standard deviation of loss fraction for species with the same functional groups.

To clarify, the critera to exclude these compounds from the analysis were added in Section 2.2.3 of the main text as following. Briefly, loss in storage were estimated, and the species with unknown loss proportion or large loss proportion higher than 50% were just used to quantify the total ROGs and were excluded in the detail speciation of ROGs.

"According to experimental results, … the loss of several alcohols (-OH) exceeded 50%. The higher polarity of alcohols is a possible reason for their larger loss proportion, as the polar species preferentially adsorbed on surface sorption sites of SUMMA canister inner walls (Batterman et al., 1998). Although volatility is another potential factor, in current experimental results for 10 days, the loss proportion has no significant dependence on volatility (Fig. S4).

For the other species such as acids (-COOH), … there were no standards used to evaluate the loss during storage. Overall, 63 specific ROGs with large uncertainty potentially (>50%) and 861 unknown masses with uncertain loss fraction were only used to quantify the total ROGs and were excluded from the further discussion of ROG composition."

**Q3:** Line 174 and 324: why is benzene chosen for normalization purposes?

**R:**

The detailed speciation of ROG emissions from residential combustion was studied comprehensively by PTR-ToF-MS combining with GC-MS/FID, while the measurement of formaldehyde and the emission factors (EFs) of ROGs were not included in our experiments. Thus, the normalization of the ROG speciation in this study was for two purposes. One was to include the

contribution of formaldehyde in the near-complete speciation of ROGs, which has been reported as one important species with considerable contribution in ROGs from residential combustion emissions (Cheng et al., 2022; Gilman et al., 2015). The other was to estimate the EFs of the newly identified species, which were rarely reported in previous studies. Both the above two purposes could be achieved by the normalization of the emission ratios of the target species (i.e. formaldehyde for the first purpose, and the newly identified species for the second purpose) with the reference species in emissions, which derived from the measurements in the present study.

Generally, all the overlapping species measured in this and the previous studies could be used for the above purpose, because the relative contribution of all the overlapping species agreed well between the current study and the previous studies as presented in Fig. S6 in the Supplementary Information. As shown in Fig. S6, the ROG species reported in different studies were various, among which aromatics were the major overlapping species in all the studies. Hence, aromatics including benzene were firstly selected as the potential reference species. Secondly, benzene was the most abundant aromatic species in the emissions of all types of fuel combustion as listed in Table S5 in the Supplementary Information, suggesting a relative lower uncertainty of the EF compared to other aromatics in previous studies. Meanwhile, the measurement of benzene in our study was also with a relative low uncertainty of 11%. Given the abovementioned, benzene was used as the reference species for the normalization.

Still, we tested other species with reported EFs and the derived EFs of ROGs with different reference species were presented in Fig. S12 in the Supplementary Information. As shown, the uncertainty of the estimated EFs of newly identified ROGs among different tests ranged from 4% (6%) to 39% (26%) for straws (coals) combustion.

We have clarified it more clearly in Section 2.3 and 3.2 in the revised manuscript, as following:

Section 2.3.4, line 235-241:

"To include the contribution of formaldehyde in the near-complete speciation of ROGs, the emission ratio of formaldehyde with the reference species in emissions was effective for this purpose. Generally, all the overlapping species measured in this and the previous studies could be used as the reference species, because the relative contribution of all the overlapping species agreed well between the current study and the previous studies as presented in Fig. S6 in the Supplementary Information. Benzene was chosen for normalization because in the emissions of all types of fuel combustion, benzene was the most abundant aromatics that were the major overlapping species between the current and previous studies."

Section 3.3, line 395-408:

"Here, benzene as well as its reported EF was used for the purpose above, as benzene was the major overlapping species in all the studies with the high abundance and a relative low uncertainty in combustion emission. We also tested other major species with reported EFs (Fig. S12). …There were no significant differences (-39%-4% for straws and 6%-26% for coals) of the estimated EFs of newly identified ROGs among different tests, which further confirmed our results were comparable with the previous studies but with more ROG species measured, as shown in Fig. S12."

**Q4:** Figure S2 (b): how many compounds were used here, and why were they chosen for comparison but not all the compounds?

**R:**

A total of 18 standard gases were used to calibrate the $H_3O^+$ PTR-ToF-MS in this study, which were all used for the comparison of calculated and measured sensitivities by $H_3O^+$ PTR-ToF-MS, as presented in Fig. S2 in the Supplementary Information.

We have clarified it more clearly in the revised Supplementary Information.

**Q5:** Given this study is based on offline analysis that some dynamic changes in emissions from residential combustion may not be reflected. Could this cause a potential bias?

**R:**

Generally, a combustion cycle mainly comprised four stages: ignition, flaming, smoldering and ember. The emissions as well as the speciation might have some dynamic changes through the process with the change of combustion conditions. The ROG composition and individual species proportion in source profiles obtained in the present study were mainly from the measurements during the flaming stage. Thus, for the present ROG composition and individual species proportion, the potential bias was related to their difference between flaming stage and the whole combustion cycle.

(1) Coal combustion

As reported in Cai et al. (2019), the emitted ROGs in the four stages accounted for 20%, 46%, 30%, and 4% of the total emissions throughout the whole cycle, respectively. The hydrocarbons which were primarily generated by pyrolysis of the volatile matter in coal, dominated the first two phases. With the increasing combustion efficiency in both the flaming and smoldering phases, the number of oxidized forms of hydrocarbons increased. The percentages of oxygenated ROGs like carbonyls and acids increased throughout the combustion cycle, while the fraction of oxygenated aromatics decreased when approaching the last stage of combustion. The fraction of N-containing ROGs increased throughout the combustion process. In summary, the composition of ROGs in each stage was shown in Fig. R2, as well as the weighted average composition during the whole cycle. The ROG composition in flaming stage and the whole cycle agreed well (Table R2), which was expected due to the small changes of ROG composition throughout the first three stages which emitted 96% of ROGs. Furthermore, by re-analyzing the data obtained from the authors, the proportion of individual species between the flaming stage and the whole cycle has a deviation varying from -50% (formic acid, $CH_2O_2$) to 22% (nitromethane, $CH_3NO_2$) for coal combustion (Fig. R4). From this point of view, for residential coal combustion, the bias of the ROG composition obtained in our study was negligible and the bias of the individual species proportion was within 50% generally.

[Figure]

**Figure R2.** Average ROG compositions for residential coal combustion at each stage and the whole cycle adapted from Cai et al. (2019) (The figure was redrawn using the data from the authors).

(2) Biomass combustion

Using the available time-resolved data of ROGs emitted from straw combustion reported by Koss et al. (2018), a re-analysis similar to coals above was conducted. The emitted ROGs in the four stages accounted for 30%, 43%, 21%, and 6% of the total emissions throughout the whole cycle, respectively. Carbonyls dominated the first three phases (41%~45%) with a decrease by ~7% during the last stage. The percentages of other oxygenated ROGs like oxygenated aromatics increased from 3% to 7% throughout the combustion cycle, while the fraction of furans (11%~14%) and esters (~1%) remained relatively stable throughout the combustion cycle. Hydrocarbons showed a similar stable contribution (8%~12%). The fraction of N-containing ROGs increased from 1% to 4% throughout the combustion process. In summary, the composition of ROGs in each stage was shown in Fig. R3, as well as the weighted average composition during the whole cycle. As presented, the ROG composition in flaming stage and the whole cycle agreed well (Table R2), which was expected due to the small changes of ROG composition throughout stage 2 and stage 3 which emitted 64% of ROGs. Furthermore, the proportion of individual species between the flaming stage and the whole cycle has a deviation ranging from -47% (formic acid, $CH_2O_2$) to 22% (benzene, $C_6H_6$) except nitromethane (109%) for straw combustion (Fig. R4). The previous study of Gilman et al. (2015) have carefully compared discrete emission ratios (ERs) during flaming and smoldering combustion and fire-integrated ERs of the whole cycle and the average slope and standard deviation of discrete versus fire-integrated ERs for select ROGs from 56 biomass burns in the US was $1.2 \pm 0.2$ (Gilman et al., 2015). From this point of view, the bias of the ROG composition of biomass combustion obtained in our study was negligible and the bias was estimated to be within 50% generally in terms of individual species.

[Figure]

**Figure R3.** Average ROG compositions for straw combustion experiments at each stage and the whole cycle redrawn using data from Koss et al. (2018) (available data from the CSD NOAA archive at https://esrl.noaa.gov/csd/groups/csd7/measurements/2016firex/FireLab/DataDownload/ (NOAA, 2018)).

[Figure]

**Figure R4.** The deviation of the proportion of individual species between the flaming stage and the whole cycle. Only the points with the upper and lower limits of deviation and outlies are marked by the species name.

**Table R2.** Comparison of average ROG compositions and the relative deviation for residential coal and straw combustion between the flame stage and the whole cycle adapted from Cai et al. (2019) and Koss et al. (2018).

| Fuel | Coal | | | Straw | | |
|---|---|---|---|---|---|---|
| Stage | Flaming | Cycle | Relative deviation | Flaming | Cycle | Relative deviation |
| Aliphatics | 21% | 20% | 5% | 10% | 9% | 11% |
| Aromatics | 35% | 33% | 6% | 1% | 1% | 0% |
| Oxygenated aromatics | 30% | 28% | 7% | 4% | 4% | 0% |
| Alcohols | 0% | 0% | 0% | 9% | 9% | 0% |
| Acids | 2% | 3% | -33% | 15% | 18% | -17% |
| Carbonyls | 6% | 7% | -14% | 45% | 42% | 5% |
| N-containing VOC | 6% | 9% | -33% | 2% | 2% | 0% |
| Furans | / | / | / | 13% | 13% | 0% |
| Esters | / | / | / | 1% | 1% | 0% |

Note: "/" means no reported data in the reference.

In summary, the bias of the fractions of species categorized by functional group from both coal and biomass combustion obtained in our study was negligible, and the bias of individual species proportion from both coal and biomass was estimated to be within 50% generally.

We have provided the discussions above in Section 3.1 in the revised manuscript as following.

Section 3.1, line 311-324:

"The ROG composition and individual species proportion in source profiles obtained in the present study were mainly from the measurements during the flaming stage. Considering the difference between flaming stage and the whole combustion cycle, the potential bias of the present results should be further discussed. By re-analyzing the data obtained from the authors (Cai et al., 2019) , the ROG composition from coal combustion in flaming stage and the whole cycle agreed well, which was expected due to the small changes of ROG composition throughout the first three stages which emitted 96% of ROGs (Fig. S9). Similar results (Fig. S10) could be concluded from the re-analysis of the reported emission data from biomass combustion by Koss et al. (2018) (Koss et al., 2018). Furthermore, the proportion of individual species between the flaming stage and the whole cycle has a deviation in the range of -50% to 22% for coal and straw combustion (Fig. S11). Actually, the previous study of Gilman et al. (2015) have carefully compared discrete emission ratios (ERs) during flaming and smoldering combustion and fire-integrated ERs of the whole cycle and the average slope and standard deviation of discrete versus fire-integrated ERs for select ROGs from 56 biomass burns in the US was $1.2 \pm 0.2$ (Gilman et al., 2015). In summary, the bias of the fractions of species categorized by functional group from both coal and biomass combustion obtained in our study was negligible, and the bias of individual species proportion from both coal and biomass was estimated to be within 50% generally."

**Q6:** Section 3.2: the SOA formation potential was estimated by using SOA yields from the literature. Are those values obtained at specific conditions? What would be the uncertainties for the estimation?

**R:**

**(1) SOA yields**

In order to estimate the SOA formation potential (SOAP) of ROGs from residential combustion, the SOA yield of observed species serving as a major parameter needs to be known. Among 80 SOA potential precursors in Table R3, the SOA yields of 44 species from previous chamber studies have been published, while SOA yields of nearly half potential precursors were still unknown.

[revised manuscript text omitted]

**Table R3.** Applied SOA yields, chamber SOA yields, references and the numbers of experiments (N).

| No. | Formula | Species | Applied Yield | Reference | Chamber Yield | N |
|-----|---------|---------|---------------|-----------|---------------|---|
| 1 | $C_2H_6$ | Ethane | / | | | |
| 2 | $C_3H_8$ | Propane | / | | | |
| 3 | $C_4H_{10}$ | Isobutane | / | | | |
| 4 | $C_4H_{10}$ | n-Butane | / | | | |

| | | | | | | | |
|---|---|---|---|---|---|---|---|
| 5 | $C_5H_{10}$ | Cyclopentane | / | | | | |
| 6 | $C_5H_{12}$ | Isopentane | / | | | | |
| 7 | $C_5H_{12}$ | n-Pentane | / | | | | |
| 8 | $C_6H_{12}$ | Methylcyclopentane | 0.017 | Assumed as Cyclohexane | | | |
| 9 | $C_6H_{12}$ | Cyclohexane | 0.017 | (Lim and Ziemann, 2009) | | 0.040 | 1 |
| 10 | $C_6H_{14}$ | 2,2-Dimethylbutane | / | | | | |
| 11 | $C_6H_{14}$ | 2,3-Dimethylbutane | / | | | | |
| 12 | $C_6H_{14}$ | 2-Methylpentane | / | | | | |
| 13 | $C_6H_{14}$ | 3-Methylpentane | / | | | | |
| 14 | $C_6H_{14}$ | n-Hexane | 0.000 | (Lim and Ziemann, 2009) | | 0.000 | 1 |
| 15 | $C_7H_{14}$ | Methylcyclohexane | 0.017 | Assumed as Cyclohexane | | | |
| 16 | $C_7H_{16}$ | 2,4-Dimethylpentane | 0.010 | Assumed as n-Heptane | | | |
| 17 | $C_7H_{16}$ | 2-Methylhexane | 0.010 | Assumed as n-Heptane | | | |
| 18 | $C_7H_{16}$ | 2,3-Dimethylpentane | 0.010 | Assumed as n-Heptane | | | |
| 19 | $C_7H_{16}$ | 3-Methylhexane | 0.010 | Assumed as n-Heptane | | | |
| 20 | $C_7H_{16}$ | n-Heptane | 0.010 | (Lim and Ziemann, 2009) | | 0.009 | 1 |
| 21 | $C_8H_{18}$ | 2,2,4-Trimethylpentane | 0.017 | Assumed as n-Octane | | | |
| 22 | $C_8H_{18}$ | 2,3,4-Trimethylpentane | 0.017 | Assumed as n-Octane | | | |
| 23 | $C_8H_{18}$ | 2-Methylheptane | 0.017 | Assumed as n-Octane | | | |
| 24 | $C_8H_{18}$ | 3-Methylheptane | 0.017 | Assumed as n-Octane | | | |
| 25 | $C_8H_{18}$ | n-Octane | 0.017 | | | 0.041 | 1 |
| 26 | $C_9H_{20}$ | n-Nonane | 0.021 | (Lim and Ziemann, 2009; Presto et al., 2010) | | 0.070 | 1 |
| 27 | $C_{10}H_{22}$ | n-Decane | 0.033 | | | 0.030-0.140 | 3 |
| 28 | $C_2H_4$ | Ethylene | / | | | | |
| 29 | $C_3H_6$ | Propylene | / | | | | |
| 30 | $C_4H_8$ | Trans-2-butene | / | | | | |
| 31 | $C_4H_8$ | 1-Butene | / | | | | |
| 32 | $C_4H_8$ | Cis-2-butene | / | | | | |
| 33 | $C_5H_{10}$ | 1-Pentene | / | | | | |
| 34 | $C_5H_{10}$ | Trans-2-pentene | / | | | | |
| 35 | $C_5H_{10}$ | Cis-2-pentene | / | | | | |
| 36 | $C_6H_{12}$ | 1-Hexene | / | | | | |
| 37 | $C_2H_2$ | Acetylene | / | | | | |
| 38 | $C_6H_6$ | Benzene | 0.096 | (Li et al., 2016a; Ng et al., 2007) | | 0.078-0.349 | 8 |
| 39 | $C_7H_8$ | Toluene | 0.200 | (Li et al., 2016a; Ng et al., 2007) | | 0.078-0.196 | 12 |
| 40 | $C_8H_8$ | Styrene | 0.016 | (Tajuelo et al., 2019) | | 0.018-0.064 | 24 |
| 41 | $C_8H_{10}$ | Ethylbenzene | 0.057 | (Li et al., 2016a) | | 0.013-0.167 | 7 |
| 42 | $C_8H_{10}$ | m/p-Xylene | 0.057 | (Li et al., 2016a; Ng et al., 2007) | | 0.035-0.154 | 44 |
| 43 | $C_8H_{10}$ | o-Xylene | 0.057 | (Li et al., 2016a) | | 0.035-0.108 | 11 |
| 44 | $C_9H_{12}$ | iso-Propylbenzene | 0.074 | (Li et al., 2016a) | | 0.031-0.110 | 4 |
| 45 | $C_9H_{12}$ | n-Propylbenzene | 0.074 | (Li et al., 2016a) | | 0.051-0.054 | 2 |
| 46 | $C_9H_{12}$ | m-Ethyltoluene | 0.074 | (Li et al., 2016a) | | 0.020-0.167 | 9 |
| 47 | $C_9H_{12}$ | p-Ethyltoluene | 0.060 | (Li et al., 2016a) | | 0.039-0.122 | 6 |
| 48 | $C_9H_{12}$ | 1,3,5-Trimethylbenzene | 0.074 | (Li et al., 2016a) | | 0.007-0.065 | 5 |
| 49 | $C_9H_{12}$ | o-Ethyltoluene | 0.074 | (Li et al., 2016a) | | 0.141-0.237 | 6 |
| 50 | $C_9H_{12}$ | 1,2,4-Trimethylbenzene | 0.074 | (Li et al., 2016a) | | 0.028-0.065 | 9 |
| 51 | $C_9H_{12}$ | 1,2,3-Trimethylbenzene | 0.074 | (Li et al., 2016a) | | 0.075-0.119 | 4 |
| 52 | $C_{10}H_{14}$ | m-Diethylbenzene | 0.048 | (Li et al., 2016a) (C10aromatics) | | 0.005-0.034 | 5 |
| 53 | $C_{10}H_{14}$ | p-Diethylbenzene | 0.048 | | | 0.005-0.034 | 5 |
| 54 | $C_5H_8$ | Isoprene | 0.440 | (Kleindienst et al., 2006; Carlton et al., 2009) | | 0.003-0.017 | 11 |
| 55 | $C_{10}H_{16}$ | alpha-Pinene | 0.346 | (Ahlberg et al., 2017) | | 0.010-0.530 | 12 |
| 56 | $C_{10}H_{16}$ | beta-Pinene | 0.346 | Assumed as alpha-Pinene | | | |
| 57 | $C_{10}H_{16}$ | Limonene | 0.346 | Assumed as alpha-Pinene | | | |
| 58 | $C_4H_6O$ | Methacrolein | / | | | | |
| 59 | $C_4H_6O$ | Methyl vinyl ketone | / | | | | |
| 60 | $C_3H_4O$ | Acrolein | / | | | | |
| 61 | $C_3H_6O$ | Propanal | / | | | | |
| 62 | $C_3H_6O$ | Acetone | / | | | | |
| 63 | $C_4H_6O$ | Crotonaldehyde | / | | | | |
| 64 | $C_4H_8O$ | n-Butanal | / | | | | |
| 65 | $C_4H_8O$ | Methyl ethyl ketone | / | | | | |
| 66 | $C_5H_{10}O$ | 2-Pentanone | / | | | | |
| 67 | $C_5H_{10}O$ | n-Pentanal | / | | | | |
| 68 | $C_5H_{10}O$ | 3-Pentanone | / | | | | |
| 69 | $C_2H_4O$ | Acetaldehyde | / | | | | |
| 70 | $CH_2O$ | Formaldehyde | / | | | | |
| 71 | $C_3H_4O_2$ | Methylglyoxal | / | | | | |
| 72 | $C_5H_4O$ | Cyclopentadienone | / | | | | |

| | | | | | | |
|---|---|---|---|---|---|---|
| 73 | $C_5H_8O$ | EVK, cyclopentanone, dihydromethylfuran | / | | | |
| 74 | $C_6H_{12}O$ | C6 carbonyls | / | | | |
| 75 | $C_7H_6O$ | Benzaldehyde | 0.200 | Assumed as toluene | | |
| 76 | $C_4H_4O$ | Furan* | 0.050 | (Gómez Alvarez et al., 2009) | 0.019-0.072 | 2 |
| 77 | $C_5H_6O$ | Methyl furan | 0.070 | (Gómez Alvarez et al., 2009) | 0.055-0.085 | 2 |
| 78 | $C_4H_4O_2$ | Furanone | 0.050 | Assumed as furan | | |
| 79 | $C_5H_4O_2$ | Furfural | 0.050 | Assumed as furan | | |
| 80 | $C_6H_8O$ | Dimethylfuran | 0.070 | Assumed as Methyl furan | | |
| 81 | $C_5H_6O_2$ | 2-Methanol furanone | / | | | |
| 82 | $C_7H_{10}O$ | TriMetfuran | 0.070 | Assumed as Methyl furan | | |
| 83 | $C_6H_8O_2$ | DiMetfuranone | 0.070 | Assumed as Methyl furan | | |
| 84 | $C_8H_6O$ | Benzofuran | 0.156 | Assumed as Styrene | | |
| 85 | $C_8H_{12}O$ | Butylfuran | 0.070 | Assumed as Methyl furan | | |
| 86 | $C_6H_6O$ | Phenol | 0.440 | (Yee et al., 2013) | 0.240-0.540 | 5 |
| 87 | $C_7H_8O$ | Cresols | 0.360 | (Henry et al., 2008) | 0.000-0.420 | 25 |
| 88 | $C_6H_6O_2$ | Benzenediols, methylfurfural | 0.370 | (Nakao et al., 2011)[b] | 0.390 | 1 |
| 89 | $C_8H_{10}O$ | C2 phenols | 0.440 | (Nakao et al., 2011)[b] | 0.130-0.730 | 7 |
| 90 | $C_7H_8O_2$ | Guaiacol, methyl benzenediols | 0.500 | (Yee et al., 2013) | 0.340-0.530 | 11 |
| 91 | $C_9H_{12}O$ | Trimethylphenol | 0.440 | Assumed as C2 phenols | | |
| 92 | $C_{10}H_8O$ | Naphthalenol | 0.440 | Assumed as C2 phenols | | |
| 93 | $C_{10}H_{12}O$ | Methyl chavicol | 0.440 | Assumed as C2 phenols | | |
| 94 | $C_8H_{10}O_3$ | Syringol | 0.370 | (Yee et al., 2013) | 0.110-0.370 | 7 |
| 95 | $C_{10}H_8$ | Naphthalene* | 0.263 | (Chan et al., 2009) | 0.190-0.300 | 5 |
| 96 | $C_{10}H_{10}$ | Dihydronaphthalene | 0.348 | Assumed as MeNap | | |
| 97 | $C_{11}H_{10}$ | Methylnaphthalene (MetNap) | 0.348 | (Chan et al., 2009) | 0.190-0.450 | 8 |
| 98 | $C_{12}H_8$ | Acenaphthalene | 0.072 | (Shakya et al., 2010) | 0.030-0.110 | 10 |
| 99 | $C_{12}H_{10}$ | Acenaphthene | 0.280 | (Shakya et al., 2010) | 0.040-0.130 | 8 |
| 100 | $C_{12}H_{12}$ | Dimethylnaphthalene (diMetNap) | 0.372 | (Chan et al., 2009) | 0.300-0.310 | 3 |
| 101 | $C_{13}H_{10}$ | Fluorene | 0.372 | Assumed as diMetNap | | |
| 102 | $C_{14}H_{10}$ | Phenanthrene, Anthracene | 0.372 | Assumed as diMetNap | | |
| 103 | $C_{16}H_{10}$ | Pyrene, Fluoranthene | 0.372 | Assumed as diMetNap | | |
| 104 | $C_8H_{18}$ | C8 Alkanes | 0.017 | | ~0.40 | 1 |
| 105 | $C_9H_{20}$ | C9 Alkanes | 0.021 | (Lim and Ziemann, 2009; Presto et al., 2010) | 0.070 | 1 |
| 106 | $C_{10}H_{22}$ | C10 Alkanes | 0.033 | | 0.010-0.150 | 3 |
| 107 | $C_{11}H_{24}$ | C11 Alkanes | 0.050 | | 0.270 | 1 |
| 108 | $C_{12}H_{26}$ | C12 Alkanes | 0.090 | (Presto et al., 2010) | ~0.01-0.09 | 8 |
| 109 | $C_{13}H_{28}$ | C13 Alkanes | 0.220 | (Presto et al., 2010)[a] | | |
| 110 | $C_{14}H_{30}$ | C14 Alkanes | 0.300 | (Presto et al., 2010)[a] | | |
| 111 | $C_{15}H_{32}$ | C15 Alkanes | 0.340 | (Presto et al., 2010) | ~0.10-0.60 | 10 |
| 112 | $C_{16}H_{34}$ | C16 Alkanes | 0.390 | (Presto et al., 2010)[a] | | |
| 113 | $C_{17}H_{36}$ | C17 Alkanes | 0.430 | (Presto et al., 2010) | 0.090-0.510 | 10 |
| 114 | $C_{18}H_{38}$ | C18 Alkanes | 0.430 | Assumed as C17 Alkanes | | |
| 115 | $C_{19}H_{40}$ | C19 Alkanes | 0.430 | Assumed as C17 Alkanes | | |
| 116 | $C_{20}H_{42}$ | C20 Alkanes | 0.430 | Assumed as C17 Alkanes | | |
| 117 | $C_{21}H_{44}$ | C21 Alkanes | 0.430 | Assumed as C17 Alkanes | | |
| 118 | $C_2H_3N$ | Acetonitrile | / | | | |
| 119 | $C_2H_5N$ | Ethenamine | / | | | |
| 120 | $C_2H_7N$ | C2 amines | / | | | |
| 121 | $C_3H_3N$ | Acrylonitrile | / | | | |
| 122 | $C_3H_5N$ | Propanenitrile | / | | | |
| 123 | $C_3H_9N$ | C3 amines | / | | | |
| 124 | $C_4H_5N$ | Pyrrole | / | | | |
| 125 | $C_4H_7N$ | Dihydropyrrole, butane | / | | | |

[a], SOA yields was estimated using the reported two-product parameters (Presto et al., 2010) which derived from the experimental yields of C12 alkanes and C17 alkanes.

[b], Only SOA yields in the absence of $NO_x$ were reviewed, which probably underestimate the SOA formation potential (Nakao et al., 2011; Yee et al., 2013).

[c], "/" in the column "Yield" denotes that these species are not the potential SOA precursors.

**Q7:** Section 3.3: The authors cited literature information to get the EFs of anthracite and straw and then applied these values to estimate the ROG emissions of residential coal and straw combustion in mainland China.

Are the quantification of EFs from limited sources representative?

**R:**

The EFs in this study were derived from the reported EF of benzene and the emission ratios (ER) of other species to benzene obtained in this study. Thus, to discuss the representative of EFs used for the estimation of emissions, two aspects should be considered as below.

(1) The representativeness of the cited EF of benzene from the literature

The major studies of the EF of benzene from residential combustion were reviewed and listed in Table R4. It can be seen the EF of benzene from each type of coal combustion generally reached a reasonable level of agreement and values of anthracite / briquette coal combustion ranged from 2 to 14 mg/kg (Cai et al., 2019; Tsai et al., 2003), which were 1-2 magnitude lower than those of bituminous coal combustion (Liu et al., 2017; Liu et al., 2015; Cai et al., 2019; Tsai et al., 2003; Wang et al., 2013). Considering the coal samples tested in this study were anthracite and briquette coal, the present study cited the latest results of anthracite coal combustion from Cai et al. (2019) study, in which the anthracite coal samples were from the two major coal production regions, i.e. Ningxia and Guizhou, and these coal types were widely used in China (Cai et al., 2019; Li et al., 2016b).

In terms of straw combustion, the EF of benzene in 12 samples from 6 literatures in total were summarized in Table R4 (Stockwell et al., 2015; Wu et al., 2022; Hatch et al., 2017; Inomata et al., 2015; Koss et al., 2018; Tsai et al., 2003) and ranged from 73 mg/kg to 800 mg/kg, which had large variations among different studies probably due to the various types of straws and combustion conditions. Among them, 7 out of 12 samples were from China straw, and their EF of benzene generally reached a reasonable level of agreement expect those of wet straw combustion. The present study used the median value of the reported EF of benzene from straw combustion in China, being of 284 mg/kg, which was derived from the simulated real-world combustion in the FLAM-4 laboratory campaign (Stockwell et al., 2015).

**Table R4.** The emission factors (EFs, mg/kg) of benzene for coals and straws reported by other studies.

| Fuel | Combustion facility | Combustion Stage | N[a] | EFs | Reference |
|---|---|---|---|---|---|
| **Coal** | | | | | |
| Anthracite coal (Ningxia/Guizhou) | commercial stove widely used in northern China | a complete burn cycle | 5 | 2-4.8 | Cai et al. (2019) |
| Briquette coal (honeycomb) | metal coal stove with/without a flue | a complete burn cycle | 3 | ~2.7-14 | Tsai et al. (2003) |
| Briquette coal | metal coal stove without flue | a complete burn cycle | 1 | 7.4 | Tsai et al. (2003) |
| Anthracite and bituminous coal | commercial stove widely used in northern China | residential burning condition | 5 | 21.5 | Wang et al. (2013) |

| Fuel | Combustion facility | Combustion Stage | Nᵃ | EFs | Reference |
|---|---|---|---|---|---|
| Bituminous coal (Shenmu/ Neimeng/ Unknown) | commercial stove widely used in northern China | a complete burn cycle | 8 | 94-156 | Cai et al. (2019) |
| Washed coal | metal coal stove with a flue | a complete burn cycle | 1 | 440 | Tsai et al. (2003) |
| Pulverized coal | metal coal stove or brick stove with a flue | a complete burn cycle | 2 | 25.8-1050 | Tsai et al. (2003) |
| Bituminous coal | commercial stove widely used in northern China | flaming or smoldering stage | 10 | 58.2-622.2 | Liu et al. (2017) |
| Bituminous coal | domestic cooking stoves | flaming or smoldering stage | / | 71-724 | Liu et al. (2015) |
| **Straw** | | | | | |
| Rice Straw (China) | a large indoor combustion room | simulated real-world conditions | 9 | 284±115 | Stockwell et al. (2015) |
| Corncob (China) | combustion chamber | simulated real-world conditions | 5 | 190 | Wu et al. (2022) |
| Rice Straw (China) | combustion chamber | mix of flaming and smoldering | 1 | 167 | Hatch & al. (2017) |
| Rice Straw (China) | a heat-resistant combustion box | flaming | 6 | 250±100 | Inomata et al. (2015) |
| Rice Straw (China) | a heat-resistant combustion box | smoldering | 8 | 400±100 | Inomata et al. (2015) |
| Rice Straw (China) | a heat-resistant combustion box | smoldering (wet fuel) | 4 | 800±500 | Inomata et al. (2015) |
| Rape Plant (China) | a heat-resistant combustion box | flaming | 2 | 220 | Inomata et al. (2015) |
| Rape Plant (China) | a heat-resistant combustion box | smoldering | 4 | 300±100 | Inomata et al. (2015) |
| Wheat Straw | a large indoor combustion room | simulated real-world conditions | 6 | 142±40 | Stockwell et al. (2015) |
| wheat residue | brick stove with a flue | a complete burn cycle | 1 | 512 | Tsai et al. (2003) |
| maize residue | brick stove with a flue | a complete burn cycle | 2 | 102-194 | Tsai et al. (2003) |
| Rice Straw | Fire Sciences Laboratory facility | mix of flaming and smoldering | 1 | 72.6 | Koss et al. (2018) |

ᵃ, The number of samples.

(2) The reasonability of the ERs obtained in the present study

The ERs of ROG species to benzene obtained in this study were used to relate the ROG EFs especially previously unmeasured or rarely measured species emissions to benzene EF. The key point of the relating above was assuming the ERs obtained in this study were consistent with those of the previous studies. The consistence could be confirmed by the good correlation (R=0.73 for straw combustion, and R=0.82 for coal combustion) of the ERs of the overlapping species between our study and the previous studies, as shown in the Fig. S6 (a) and (e) in the Supplementary Information presented the correlation.

We have provided the discussions in the revised manuscript as following and in the Supplementary

Information:

Section 3.3, line 397-405:

"The major studies of the EF of benzene from residential combustion were further reviewed and listed in Table S4. Considering the coal samples tested in this study were anthracite and briquette coal, the present study cited the latest reported EF of benzene from anthracite coal combustion by Cai et al. (2019), which agreed with the other reported values of anthracite / briquette coal combustion (Tsai et al., 2003) and 1-2 magnitude lower than those of bituminous coal combustion (Liu et al., 2017; Liu et al., 2015; Cai et al., 2019; Tsai et al., 2003; Wang et al., 2013). In terms of straw combustion, the present study used the median value of the reported EF of benzene from straw combustion in China, being of 284 mg/kg, which was derived from the simulated real-world combustion in the FLAM-4 laboratory campaign (Stockwell et al., 2015). More particular consideration about selection of the reported EF of benzene was described in the Supplementing Information."

Section 2.3.4, line 408-414:

"To relate the ROG EFs especially previously unmeasured or rarely measured species emissions to benzene EF, the ER of ROG species to benzene was the ratio of their concentrations in the sample, and the average ER in different samples of each type of fuel was used in this study, as listed in Table S5. The key point of the relating above was assuming the ERs obtained in this study were consistent with those of the previous studies. The consistence could be confirmed by the good correlation (R=0.73 for straw combustion, and R=0.82 for coal combustion) of the ERs of the overlapping species between our study and the previous studies, as shown in the Fig. S6 (a) and (e) in the Supplementary Information presented the correlation."

Is anthracite representative of residential coal combustion in mainland China?

**R:**

Through an extensive literature search, little statistical data about the consumption of bituminous and anthracite coal in the residential sector of China was founded. From the rural energy survey in 2013-2014, the raw coal contributed 97% and 55% of the residential coal consumptions (raw coal and honeycomb briquette) in Baoding (Zhi et al., 2017; Zhi et al., 2015) and Beijing (Zhao et al., 2015), respectively. Assuming that the raw coal is equivalent to bituminous coal, the proportions of bituminous coal can be obtained roughly (Cai et al., 2019). China has been carrying out toughest-ever clean energy substitution and vigorously replacing bituminous coal with anthracite in response to the clean action plan in the residential sector since 2013, which were further strengthened from 2017 to 2020 during the three-year battle against air pollution. National Energy Administration strictly prohibit the sale of low rank coal in Action Plan for Clean and Efficient Utilization of Coal (2015-2020) (http://zfxxgk.nea.gov.cn/). The use and sale of bituminous coal were generally not allowed (Luo, 2019).

Thus, we could expect the large decrease of the use of bituminous coal in residential sector in China although there is no updated statistical data appliable. This study assumed that anthracite is the main residential coal type to roughly estimate ROG emissions in China. Even if the emission factors of

bituminous coal were applied, the ROG emissions from residential coal combustion in China would increase by approximately 1-2 orders of magnitude, which were still lower than the ROG emissions from biomass combustion. More refined energy consumption statistics are necessary to update as adjustment of Chinese energy structure, which is beyond the scope of our effort.

We have stressed the assumption in Section 3.3 and Fig. 5 in the revised manuscript as following:

"Notably, the appliable data about the contribution of bituminous and anthracite coal were from the rural energy survey conducted about ten years ago (2013-2014), which indicated the bituminous coal contributed 97% and 55% of the residential coal consumptions in Baoding (Zhi et al., 2017; Zhi et al., 2015) and Beijing (Zhao et al., 2015). China has been carrying out toughest-ever clean energy substitution and vigorously replacing bituminous coal with anthracite in response to the cleaning action plan in the residential sector since 2013, which were further strengthened from 2017 to 2020 during the three-year battle against air pollution (eg. Action Plan for Clean and Efficient Utilization of Coal (2015-2020), http://zfxxgk.nea.gov.cn/). The use and sale of bituminous coal were generally not allowed (Luo, 2019). Thus, we could expect the large decrease of the use of bituminous coal in residential sector in China although there is no updated statistical data appliable. This study assumed that anthracite is the main residential coal type to roughly estimate ROG emissions in China.

…Even if the emission factors of bituminous coal were applied, the ROG emissions from residential coal combustion in China would increase by approximately 1-2 orders of magnitude, which were still lower than the ROG emissions from biomass combustion. More refined energy consumption statistics are necessary to update as adjustment of Chinese energy structure."

[revised manuscript text omitted]